# CONSISTENCY REGULARIZATION FOR DOMAIN GENERALIZATION WITH LOGIT ATTRIBUTION MATCHING

## ABSTRACT

Domain generalization (DG) is about training models that generalize well to unseen domains with different data distributions than the training domains. It has recently been shown that an effective way to achieve good out-of-distribution (OOD) performance is targeted data augmentation, which randomizes spurious factors while preserving robustly predictive factors in training examples. Data augmentation (DA) naturally leads to paired training examples that share the same semantic contents, which can be utilized via consistency regularization (CR). In this paper, we show that CR can further boost OOD performance on top of targeted DA. We also propose a novel CR-based DG method called Logit Attribution Matching (LAM). In comparison with previous CR-based DG methods, a key advantage of LAM is that it leverages class labels often associated with semantic sharing pairs. Empirically we find that LAM consistently outperforms previous CR-based DG methods on benchmarks with multiple classes. It is the only one that can consistently improve over the targeted DA on all the datasets tested in our experiments. To justify the CR-based approach to DG theoretically, we establish conditions for optimal DG in a causal framework and explain how CR is related to those conditions.

## 1 INTRODUCTION

Deep learning models are successful under the independent and identically distributed (i.i.d.) assumption that test data are drawn from the same distribution as that of training data. However, models that generalize well in-distribution (ID) may be generalizing in unintended ways out-of-distribution (OOD) (Szegedy et al., 2013; Shah et al., 2020; Geirhos et al., 2020; Di Langosco et al., 2022). Some image classifiers with great ID performance, in fact, rely on background cues to predict the class of foreground objects, leading to poor OOD performance (Beery et al., 2018; Zech et al., 2018; Xiao et al., 2020). Such reliance on spurious correlations is subject to various kinds of domain shift, affecting many real-world applications where the i.i.d. assumption cannot be guaranteed (Michaelis et al., 2019; Alcorn et al., 2019; Koh et al., 2020; Ali et al., 2022).

*Domain generalization* (DG) deals with the conundrum of generalizing under domain shift. It is difficult because actual test domains may vary from the training domains in numerous ways. *Multi-source DG* methods tackle this issue by utilizing data from multiple training domains, however, as Gulrajani & Lopez-Paz (2021) has shown, these methods are not generally better than the classic Empirical Risk Minimization (ERM) (Vapnik, 1998). The stagnation of algorithmic improvement under the multi-source setting suggests that we may need to rethink what kind of data to use and how to use them to improve OOD performance.

A promising alternative to multi-source DG is *data augmentation* (DA). By generating more data from existing ones, DA exposes a model to more feature variations during training and thereby enhances its capability in dealing with novel domains. Generic DA methods such as Mixup (Zhang et al., 2017) and RandAugment (Cubuk et al., 2020) have been shown to improve OOD performance in many cases, although the improvement is inconsistent across datasets. The limitation of these methods is that their efficacy largely depends on how well the induced variation aligns with the domain shift (Wiles et al., 2022). To address this, Gao et al. (2023) proposed *targeted augmentation* (or *targeted DA*) which promotes the idea of designing specific data augmentation tailored for each individual task. In essence, targeted DA serves as a medium for transferring vital domain knowledge

from human to machine that is practically infeasible or inefficient through generic DA and multiple training domains. Empirical evidence shows targeted DA outperforming previous generic DA and multi-source baselines significantly on two real-world datasets in WILDS (Gao et al., 2023).

In this paper, we further investigate the potential of DA by studying its synergy with *consistency regularization* (CR), a widely used technique in machine learning that encourages a model to make similar predictions on similar inputs (Bachman et al., 2014; Zhang et al., 2020; Chen et al., 2020; Caron et al., 2021). The rationale behind this synergy is that DA naturally leads to paired training examples that share the same semantic content, which can be further utilized via CR to enhance DG. Currently, there is a small number of DG methods that are based on CR (Hendrycks et al., 2019; Mitrovic et al., 2021; Heinze-Deml & Meinshausen, 2021; Mahajan et al., 2021; Robey et al., 2021; Ouyang et al., 2021; Wang et al., 2022). Those methods involve several ways to create pairs (or groups) of semantic sharing examples and various ways to enforce CR between the examples.

We make three contributions in this paper. *First, we provide a systematic empirical evaluation of multiple CR-based DG methods on a range of tasks.* This is the first time they are evaluated together on such a scale and rigor under the same fair setting. We concretely show that CR, if done properly, can improve OOD performance when combined with DA, especially with targeted DA, in addition to the performance gains already achieved by DA itself. *Second, we provide theoretical insights on the performance gains.* We start by establishing a set of sufficient conditions for optimal DG in a causal framework, and then show that CR can be viewed as natural way to achieve a key condition for optimal DG using semantic sharing (SS) pairs, namely causal-invariant prediction.

*Third, we propose a novel CR method called logit attribution matching (LAM).* When a training example is augmented, it is usually done in a way to preserve the information for a target class, resulting in a labeled SS pair. Previous CR-based DG methods ignore the class label and aim to match the entire feature/logit/probability vectors of the pair. This might adversely affect the latent features of other classes. LAM alleviates this problem by focusing only on the target class, encouraging equal latent features for both examples in a SS pair. Empirical results show that LAM consistently outperforms other CR-based DG methods and other representative DG methods on benchmarks with multiple classes. It is the only one that can consistently improve over targeted DA on all datasets tested in our experiments. The previous best reported OOD performances of targeted DA on two of the datasets, IWILDCAM2020-WILDS and CAMELYON17-WILDS, are 36.5% and 90.5% (Gao et al., 2023) respectively. With LAM, we achieved 41.2% and 93.5% using the same architectures.

## 2 RELATED WORK

**DOMAIN GENERALIZATION**: Domain generalization (DG) is a fundamental problem in machine learning and has attracted much attention in recent years. A large number of methods have been proposed. In this section, we briefly review several representative methods that are frequently used as baselines in the literature. They are also used in our experiments as baselines. Most DG methods assume multiple training domains. Among those multi-source methods, Group Distributionally Robust Optimization (GDRO) (Sagawa et al., 2020) seeks to minimize the worst-case risk across all possible training domains. Invariant Risk Minimization (IRM) (Arjovsky et al., 2019) regularizes ERM with a penalty that enforces cross-domain optimality on the classifier. Variance Risk Extrapolation (V-REx) (Krueger et al., 2020) penalizes the variance of risks in different training domains. Domain-Adversarial Neural Networks (DANN) (Ganin et al., 2016) aims at mapping inputs from each training domain to an invariant distribution in the feature space from which the original domain is indistinguishable. Single-source DG does not assume access to multiple training domains. One of the main approaches to single-source DG is to discover predictive features that are more sophisticated than simple cues spuriously correlated with labels. Representation Self-Challenging (RSC) (Huang et al., 2020) and Spectral Decoupling (SD) (Pezeshki et al., 2021) are two prominent methods in this direction. SD suppresses strong dependencies of output on dominant features by regularizing the logits. RSC aims to achieve the same goal in a heuristic manner. At each iteration of training, it mutes the feature units associated with the highest gradients, such that the network is forced to predict the labels through other less salient features.

**CONSISTENCY REGULARIZATION**: Consistency regularization (CR) encourages a model to make similar predictions on similar inputs. The idea originated from the semi-supervised learning literature (Bachman et al., 2014; Sohn et al., 2020). It is also used in contrastive learning (Chen

et al., 2020) and non-contrastive self-supervised learning (Caron et al., 2021). There is a small number of DG methods based on CR (Hendrycks et al., 2019; Mitrovic et al., 2021; Heinze-Deml & Meinshausen, 2021; Mahajan et al., 2021; Robey et al., 2021; Wang et al., 2022). They differ in their ways to create and use SS pairs.

A straightforward way to create SS pairs is to use generic DA methods such as CutMix (Yun et al., 2019) and RandAugment (Cubuk et al., 2020), or Targeted Augmentations (Gao et al., 2023) that are specifically designed for DG. An input image $x$ and an augmentation $\tilde{x}$ form an SS pair. SS pairs can also be created/obtained in ways other than conventional DA. When analyzing the CelebA dataset (Liu et al., 2015), Heinze-Deml & Meinshausen (2021) pair up photos of the same person. When analyzing medical images, Ouyang et al. (2022) create pairs by performing image transformations to simulate different possible acquisition processes. In the case of multiple source domains, SS pairs can be learned. Robey et al. (2021) and Wang et al. (2022) build image-to-image translation networks between domains and use them to create pairs. Mahajan et al. (2021) propose an iterative algorithm that uses contrastive learning to map images to a latent space, and then match up images from different domains that have the same class label and are close to each other in the latent space.

Let $x$ and $\tilde{x}$ be two training examples with the same semantic content. There are a number of ways to use them to regularize a model: *Probability matching* minimizes the divergence between the output distributions $P(\hat{Y}|x)$ and $P(\hat{Y}|\tilde{x})$ (Hendrycks et al., 2019; Mitrovic et al., 2021); logit matching minimizes the difference between the logit vectors $logit(x)$ and $logit(\tilde{x})$ (Heinze-Deml & Meinshausen, 2021); and *feature matching* minimizes the difference between the feature vectors $f(x)$ and $f(\tilde{x})$ (Mahajan et al., 2021).

## 3 A CAUSAL THEORY OF DOMAIN GENERALIZATION

In DG, a *domain $d$* is defined by a distribution $P^d(X, Y)$ over the space of input-label pairs $(X, Y)$. We assume the pairs are generated by the casual model shown in Figure 1 (a): (1) the input of a training example $X$ is generated from two latent variables $X^c$ and $X^n$, (2) $X^c$ and $X^n$ are statistically correlated, and (3) the label $Y$ is generated from only $X^c$. The *causal mechanisms* that generate $X$ and $Y$ are assumed to be invariant across domains. The corresponding conditional distributions are denoted as $P^*(X|X^c, X^n)$ and $P^*(Y|X^c)$. The joint distribution $P^d(X^n, X^c)$ of the two latent variables may change across domains. The variable $X^c$ denotes the essential information in an image $X$ that a human relies on to assign a label $Y$ to the image. In contrast, the variable $X^n$ denotes the other aspects of $X$ that are not essential to label assignment. Similar models are also proposed by Liu et al. (2021); Heinze-Deml & Meinshausen (2021); Mahajan et al. (2021); Mitrovic et al. (2021); Ye et al. (2022), where $X^c/X^n$ are referred to as semantic/variation factors, causal/non-causal factors, content/style, etc. In this paper we will call them *core factors* and *non-core factors* respectively. Moreover, we refer to the model as the *causal latent decomposition (CLD) model*.

To ground the CLD model, we need to specify three distributions: $P^d(X^c, X^n)$, $P^*(X|X^c, X^n)$ and $P^*(Y|X^c)$. Together, the three distributions define a joint distribution over the four variables:

$$P^d(X^c, X^n, X, Y) = P^d(X^c, X^n)P^*(X|X^c, X^n)P^*(Y|X^c).$$

This joint distribution defines a domain in the CLD framework. We refer to the collection of all such domains for some fixed $P^*(X|X^c, X^n)$ and $P^*(Y|X^c)$ as a *CLD family*.

Let $\mathscr{X}^c$ and $\mathscr{X}^n$ be the sets of all possible values of the latent variables $X^c$ and $X^n$ respectively. Consider an example $x$ generated by $P^*(X|x^c, x^n)$ from a pair of values $(x^c, x^n) \in \mathscr{X}^c \times \mathscr{X}^n$ of $X^c$ and $X^n$.[1] Let $\tilde{x}$ be another example sampled from the same $x^c$ and a different $\tilde{x}^n$. Formally,

$$x \sim P^*(X|x^c, x^n), \tilde{x} \sim P^*(X|x^c, \tilde{x}^n). \tag{1}$$

The two examples $x$ and $\tilde{x}$ contain the same semantic contents and hence should be classified into the same class. In this sense, $x$ and $\tilde{x}$ make up a *semantic sharing (SS) pair*. Let $\hat{P}_\theta(\hat{Y}|X)$ be a prediction model with parameters $\theta$. It is said to be *causal-invariant* if

$$\hat{P}_\theta(\hat{Y}|x) = \hat{P}_\theta(\hat{Y}|\tilde{x}), \tag{2}$$

---

[1]We use upper case letters to denote variables and lower case letters to denote their values.

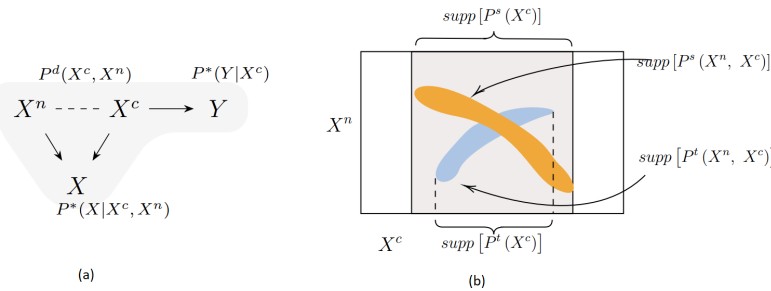

Figure 1: A causal model for DG (a) and an illustration of conditions for optimal DG (b).

for any $x$ and $\tilde{x}$ sampled via (1) from any $x^c \in \mathscr{X}^c$, $x^n \in \mathscr{X}^n$ and $\tilde{x}^n \in \mathscr{X}^n$. In other words, the prediction output does not change in response to variations in the non-core factors $X^n$ as long as the core factors $X^c$ remain fixed.

The concept of *causal-invariant prediction (CIP)*, as defined above, is not a new one and frequently appears in the DG literature in various forms (Mitrovic et al., 2021; Mahajan et al., 2021). It is closely related to a notion described in Peters et al. (2016) that has a very similar name, *invariant causal prediction (ICP)*. One difference between the two is that we are concerned with the invariance of prediction with respect to a latent variable $X^n$, which is entangled with another latent variable $X^c$ to yield the observed input $X$. Our objective is to make the prediction model invariant to $X^n$, although we cannot observe it. In contrast, ICP is concerned with the invariance of prediction with respect to observed variables and does not address the problem of entanglement.

**Theorem 1  (Conditions for Optimal DG)** *Let $\hat{P}_\theta$ be a prediction model for a CLD family, and $P^s$ and $P^t$ be a source and a target domains from the family respectively. Suppose: 1). $\hat{P}_\theta$ minimizes the in-distribution (ID) cross entropy loss $\ell_{P^s}(\hat{P}_\theta) = \mathbb{E}_{(x,y)\sim P^s(X,Y)}[-\log \hat{P}_\theta(\hat{Y} = y|x)]$; 2). $\hat{P}_\theta$ is causal-invariant; and 3). $\mathrm{supp}[P^t(X^c)] \subseteq \mathrm{supp}[P^s(X^c)]$. Then, the prediction model $\hat{P}_\theta$ also minimizes the out-of-distribution (OOD) cross entropy loss $\ell_{P^t}(\hat{P}_\theta) = \mathbb{E}_{(x,y)\sim P^t(X,Y)}[-\log \hat{P}_\theta(\hat{Y} = y|x)]$. In other words, it generalizes optimally to the target domain.*

The proof of this theorem can be found in Appendix A. Note that the first condition requires $\ell_{P^s}(\hat{P}_\theta)$ be minimized not only with respect to the parameters of the prediction model $\hat{P}_\theta$, but also its architecture. The support $\mathrm{supp}[P^d(X^c)] := \{x^c | P^d(x^c) > 0\}$ of $P^d(X^c)$ consists of all core factors that appear in a domain $P^d$. To gain an intuitive understanding of Theorem 1, take a look at Figure 1 (b). Training examples $x$ are sampled from a latent space spanned by the values $x^c$ and $x^n$ of the latent variables $X^c$ and $X^n$, which we depict as a two-dimensional box. A prediction model is causal-invariant if it makes the same prediction for examples sampled from the same "vertical line" in the latent space. If such a causal-invariant model also minimizes the cross-entropy loss of a source domain, then it makes optimal prediction on all examples $\tilde{x}$ sampled from $\mathrm{supp}[P^s(X^c)] \times \mathscr{X}^n$ (the inner rectangle), not only those sampled from $\mathrm{supp}[P^s(X^c, X^n)]$. This enables optimal generalization to any target domain $P^t$ such that $\mathrm{supp}[P^t(X^c)] \subseteq \mathrm{supp}[P^s(X^c)]$.

Theorem 1 is similar in spirit to the theoretical findings of several previous works (Mahajan et al., 2021; Mitrovic et al., 2021; Wang et al., 2022). However, there is a crucial distinction in that the prior results are linked to particular methods for achieving the conditions of optimal domain generalization. In contrast, our theorem solely concentrates on why these conditions result in optimal domain generalization. This allows us to use it to motivate the consistency regularization approach to DG, which will be shown in the next section.

## 4  CONSISTENCY REGULARIZATION FOR DOMAIN GENERALIZATION

Suppose we have a collection of SS pairs $\Pi = \{(x_k, \tilde{x}_k)\}_{k=1}^K$. To achieve the first two conditions for optimal DG, a straightforward approach is to solve the following constrained optimization problem:

$$\min_\theta \quad \mathbb{E}_{(x,y)\sim P^s}[-\log \hat{P}_\theta(\hat{Y} = y|X = x)]$$

$$\text{subject to} \quad \hat{P}_\theta(\hat{Y}|X = x_k) = \hat{P}_\theta(\hat{Y}|X = \tilde{x}_k) \quad \forall (x_k, \tilde{x}_k) \in \Pi.$$

If we turn the equality constraints into a *consistency regularization* term, the problem becomes:

$$\min_{\theta} \quad \mathbb{E}_{(x,y)\sim P^s}[-\log \hat{P}_{\theta}(\hat{Y}=y|X=x)] + \lambda\frac{1}{K}\sum_{k=1}^{K} r_{\theta}(x_k, \tilde{x}_k), \tag{3}$$

where $\lambda$ is a balancing parameter.

### 4.1 Previous CR-based DG methods

Suppose $\hat{P}_{\theta}$ consists of a feature extractor $f_{\phi}$ with parameters $\phi$ and a linear classification head $g_{\mathbf{w}}$ with parameters $\mathbf{w}$. Hence, $\theta = (\phi, \mathbf{w})$. For an input $x$, let $f_{\phi}^u(x)$ be the component of the feature vector $f_{\phi}(x)$ that is associated with a feature unit $u$. Let $w_{uy}$ be the weight between the feature unit $u$ and the output unit for a class $y$. The logit for the class $y$ is $z_{\theta}^y(x) = \sum_u w_{uy} f_{\phi}^u(x)$. [2]

For each SS pair, the regularization term $r_{\theta}(x_k, \tilde{x}_k)$ can be defined in several ways:

- *Probability Matching (KL)*: $r_{\theta}(x_k, \tilde{x}_k) = KL(\hat{P}_{\theta}(\hat{Y}|x_k)||\hat{P}_{\theta}(\hat{Y}|\tilde{x}_k))$,
- *Probability Matching (JS)*: $r_{\theta}(x_k, \tilde{x}_k) = JS(\hat{P}_{\theta}(\hat{Y}|x_k)||\hat{P}_{\theta}(\hat{Y}|\tilde{x}_k))$,
- *Logit Matching*: $r_{\theta}(x_k, \tilde{x}_k) = \sum_y (z_{\theta}^y(x_k) - z_{\theta}^y(\tilde{x}_k))^2$,
- *Feature Matching*: $r_{\theta}(x_k, \tilde{x}_k) = \sum_u (f_{\phi}^u(x_k) - f_{\phi}^u(\tilde{x}_k))^2$.

Probability matching (KL) is used in Representation Learning via Invariant Causal Mechanisms (ReLIC) (Mitrovic et al., 2021); probability matching (JS) is used in *AugMix* (Hendrycks et al., 2019); logit matching is used in Conditional Variance Regularization (CoRE) (Heinze-Deml & Meinshausen, 2021); and feature matching is used in MatchDG (Mahajan et al., 2021). It is worth noting that while we focus on pairs for simplicity, logit and feature matching can also be applied to the case with groups of multiple examples that share the same semantic contents. To achieve this, we can simply replace the sum of squared differences with the sum of variances. This is done in CoRE and MatchDG.

An SS pair $(x_k, \tilde{x}_k)$ is usually obtained by changing an original training example $x_k$. The semantic information for the ground-truth label $y_k$ is typically preserved. The simplest way to use the augmented example $(\tilde{x}_k, y_k)$ is to add it to the training set. As a method to improve OOD performance, this is known as data augmentation (DA). The effect is to push both $P_{\theta}(\hat{y}=y_k|x_k)$ and $P_{\theta}(\hat{y}=y_k|\tilde{x}_k)$ toward 1.0. When $P_{\theta}(\hat{y}=y_k|x_k)$ and $P_{\theta}(\hat{y}=y_k|\tilde{x}_k)$ differ significantly from each other, CR places additional regularization force on the model. This is why CR can further improve OOD performance on top of DA.

Among the aforementioned CR-based DG methods, probability matching aims to directly enforce the equality (2) on the SS pairs. This makes the model more causal-invariant, which is conducive to good OOD performance according to Theorem 1. Feature matching aims to make the feature vectors of $x_k$ and $\tilde{x}_k$ identical, which is a stronger condition than the equality (2). The same can be said about logit matching. Enforcing a condition stronger than necessary can hurt OOD performance.

### 4.2 Logit Attribution Matching

As mentioned above, the semantic information for the ground-truth label $y_k$ is typically preserved when an training example $x_k$ is changed to produce an augmented example $\tilde{x}_k$. This means that our SS pair is actually labelled and can be written as $(x_k, \tilde{x}_k : y_k)$. The crucial difference between labeled and unlabeled SS pairs, used in most prior CR-based DG methods, lies in the information they carry: while unlabelled pairs imply that $x_k$ and $\tilde{x}_k$ hold identical information about all classes, labelled pairs signify that they contain the same information about one specific class $y_k$. This distinction matters because, in practice, it is much harder to ensure that $x_k$ and $\tilde{x}_k$ hold identical information about all classes of interest than a single class $y_k$, especially when the total number of classes is large. For instance, images labeled with class $y_k$ may have small objects of other classes in the background, a type of label noise common in machine learning datasets (Northcutt et al., 2021).

---

[2] Assume the bias is represented by a dummy unit.

Therefore, in real-world scenarios, rather than matching the probability/logit distributions over all classes or the entire feature vectors, it is more logical to match only the features pertinent to the $y_k$ class. The narrowed focus on the $y_k$ reduces the theoretical requirement from Theorem 1 for pairs to perfectly share the same features across all classes of interest. With this perspective, we propose a new regularization term to better leverage the information in the labelled SS pairs $(x_k, \tilde{x}_k : y_k)$:

$$r_\theta(x_k, \tilde{x}_k) = \sum_u w_{uy_k}^2 (f_\phi^u(x_k) - f_\phi^u(\tilde{x}_k))^2, \tag{4}$$

where the $w_{uy_k}$ is the weight in the classification layer (assumed to be linear, so effectively the last layer) between the feature unit $u$ and the logit of class $y_k$. To understand the underlying idea of the proposed regularization term, imagine the ideal case where the feature extractor $f$ cleanly disentangles $x_k^c$ and $x_k^n$, in the sense that the feature vector $f_\phi(x_k)$ is divided into two parts $H_1(x_k)$ and $H_2(x_k)$ that depend only on $x_k^c$ and $x_k^n$ respectively, with $H_1(x_k) = H_1(\tilde{x}_k)$ and $H_2(x_k) \neq H_2(\tilde{x}_k)$. A causal-invariant predictor should predict $y_k$ based only on $H_1(x_k)$. Hence the weights $w_{uy_k}$ for the feature units $u$ in $H_2$ should be 0, such that the features from the non-core factors $x_k^n$ are ignored. In practice, it is difficult to cleanly disentangle $x_k^c$ and $x_k^n$. In such a case, our regularization term has the following effects:

1) It encourages $g_\mathbf{w}$ to put high weights $w_{uy_k}$ on the units $u$ where $f_\phi^u(x_k) \approx f_\phi^u(\tilde{x}_k)$, and

2) It encourages $f_\phi$ to make $f_\theta^u(x_k) \approx f_\phi^u(\tilde{x}_k)$ for units $u$ with high weights $w_{uy_k}$.

Note that $\sum_u w_{uy_k}^2 (f_\phi^u(x_k) - f_\phi^u(\tilde{x}_k))^2 = \sum_u (w_{uy_k} f_\phi^u(x_k) - w_{uy_k} f_\phi^u(\tilde{x}_k))^2$. The regularization term essentially matches the contributions $w_{uy_k} f_\phi^u(x_k)$ and $w_{uy_k} f_\phi^u(\tilde{x}_k)$ from the feature units $u$ to the logit $z_\theta^{y_k}$ of $y_k$. As such, we call the new regularization term *logit attribution matching (LAM)*. It encourages the logits of $y_k$ for $x_k$ and $\tilde{x}_k$ be computed from the same latent features and using the same weights. This is a more fine-grained regularization force than equation (2), prompting the model to concentrate on the specific features that are directly associated with the specific class $y_k$.

## 5 EXPERIMENTS

There are four objectives in our empirical studies. First, we determine the effectiveness of previous CR-based DG methods in improving OOD performance on top of targeted DA, and their relative merits. Those methods were proposed and evaluated separately and with different SS pair creation methods (see Section 2). This is the first time they are evaluated together on the same data, with SS pairs created by targeted DA. Second, we compare LAM with previous CR-based DG methods. Third, we compare LAM with other representative DG methods. Finally, we investigate the impact of the quality and quantity of SS pairs on the performance of LAM, and demonstrate that LAM helps models focus on core factors for prediction.

### 5.1 DATASETS

Our experiments involve five datasets, three with background shifts and two with style shifts.

**IWILDCAM2020-WILDS (iWildCam)** (Koh et al., 2020) consists of camera trap photos of animals taken at different locations. There are totally 217,640 images and 182 classes. The training domain of iWildCam comprises images from 200 locations, while the test domain contains images from a separate set of locations. A validation domain with images from additional locations is also provided for model selection and hyperparameter tuning. For generating the augmented examples for iWildCam, we cut-and-paste the animals in the training image to another image without animals taken at a different location where the same animals sometimes appear. This is believed to randomize the spurious low-level background factors while preserving the robustly predictive species and habitat factors in the training examples.

**ImageNet-9** (Xiao et al., 2020) consists of about 50,000 images from ImageNet, and it involves nine coarse-grain classes. Several synthetic variations are created by segmenting the foreground of each image and place it onto a different background. In our main experiments, we use the variations where the segmentation is done using GrabCut (Rother et al., 2004). The synthetic images with a

black background are used as augmented examples. The test domain *mixed-rand* consists of samples where the foreground of an original image is placed onto the background of a random image.

**NICO** (He et al., 2020) includes around 25,000 images across 19 classes of animals or vehicles captured in different contexts such as "at home" or "on the beach". Each class encompasses 9 or 10 different contexts. As there is no predefined train-test split, we randomly select one context per class for testing and use the remaining contexts for training. Similar to ImageNet-9, we perform augmentation on NICO using the GrabCut segmentation. We place the foreground segmentation onto the background of a random image.

**CAMELYON17-WILDS (Camelyon)** contains histopathology images from multiple hospitals for binary tumor classification. Images from three hospitals are used for training, while images from a fourth and fifth hospital are used for testing and validation respectively. There are stylistic variations among images from different hospitals. One key stylistic difference often observed is the stain color. Therefore, the stain color jitter (Tellez et al., 2018) is applied to training images to create augmented examples. The jitter can effectively randomize the average stain level of images.

**PACS** (Li et al., 2017) contains images of objects and creatures depicted in four different style domains: *photo*, *art*, *cartoon* and *sketch*. In total, it includes 9,991 images of 7 classes. Following the common practice (Li et al., 2017; Gulrajani & Lopez-Paz, 2021), we train three separate models, using the *art*, *cartoon* and *sketch* domain respectively as the test domain, while the remaining three domains are used for training in each model. We report the average accuracy of the three models for each DG method. The *photo* domain is excluded from being the test domain, since we perform augmentation on images from it. Specifically, we employ StableDiffusion (Rombach et al., 2022) to transform the style of images from the *photo* domain, guided by the text prompt "a minimalist drawing of a `class_name`, outline only, no texture" (more details in Appendix B).

For the iWildCam and Camelyon, whose shifts arise naturally from the real-world application, we use all training images including animals for iWildCam, and all training images for Camelyon to create the augmented examples. To assess the performance of CR-based DG methods when only a relatively small proportion of augmented examples is available, we only apply augmentation on approximately 5% of the training data for ImageNet-9 and NICO, and about 10% of the training data for PACS. All CR-based methods have a balancing parameter $\lambda$, which is tuned on the validation domain for iWildCam and Camelyon, and tuned on the training domain for the other three datasets.

## 5.2 PRETRAINED MODELS

In our experiments, we started with pretrained models and finetuned them on the aforementioned datasets. Different pretained models are used for different datasets so as to be consistent with previous work (Gao et al., 2023). Specifically, we used the ImageNet pretrained ResNet-50 model on iWildCam, and a randomly initialized DenseNet-121 model (Huang et al., 2017) on Camelyon. We used the CLIP-pretrained (Radford et al., 2021) ViT-B/16 model (Dosovitskiy et al., 2020) on ImageNet-9 and NICO, and the CLIP-pretrained ResNet-50 model on the PACS experiment. See Appendix D for more details.

## 5.3 PERFORMANCES OF PREVIOUS CR-BASED DG METHODS

Our main results are summarized in Table 1. The first row shows the OOD performances of the models obtained by minimizing the empirical cross entropy (ERM). The second row shows the performance of the models obtained via ERM with the target-augmented data added to the training set (ERM+DA). We see that targeted DA improves OOD performances drastically on iWildCam and Camelyon. Those results are consistent with what were reported in Gao et al. (2023). Targeted DA also boosts OOD performances on ImageNet-9, NICO and PACS albeit to lesser extents.

The four rows in the middle show the results for previous CR-based DG methods. We see that the best of them improves over ERM+DA significantly on iWildCam and Camelyon, and slightly on ImageNet-9 and PACS. Those indicate that CR, if done properly, can indeed improve OOD preformance when combined with targeted DA.

None of the four methods dominates the others across the board. Probability matching generally outperforms logit and feature matching, particularly on iWildCam. We conjecture that this is because

Table 1: OOD performances of models trained using ERM, ERM+DA, previous CR-Based DG methods, and LAM. The OOD performance of a model is assessed on held-out test domain(s) using Macro F1 score on iWildCam or classification accuracy on all other datasets. Each model is trained three times, with the standard deviation reported. The best result is in **bold** and second best is underlined. Arrows indicate changes relative to ERM+DA. The *Count* ↓ represents the number of datasets on which the CR-based DG method is worse than ERM+DA: a lower count indicates better performance relative to ERM+DA.

| | ImageNet-9 | NICO | PACS | iWildCam | Camelyon | iWildCam-N | Count ↓ |
|---|---|---|---|---|---|---|---|
| ERM | 83.3±1.1 | 95.3±0.1 | 82.8±0.5 | 30.2±0.3 | 65.2±2.6 | 15.8±2.1 | — |
| ERM+DA | 86.0±1.0 | 95.9±0.3 | 84.5±0.5 | 36.5±0.4 | 90.5±0.9 | 28.2±0.5 | — |
| Prob. (JS) | 86.0±0.4 − | 95.0±0.3 ↓ | 84.3±0.3 ↓ | 37.1±0.4 ↑ | **94.8±1.2** ↑ | 25.5±0.6 ↓ | 3 |
| Prob. (KL) | 86.9±0.2 ↑ | 95.4±0.2 ↓ | 85.0±1.0 ↑ | 40.3±0.3 ↑ | 92.8±1.5 ↑ | 26.3±0.7 ↓ | 2 |
| Logit | 86.8±0.6 ↑ | 95.3±0.2 ↓ | 83.1±0.8 ↓ | 34.3±0.5 ↓ | 93.4±0.3 ↑ | 23.9±0.5 ↓ | 4 |
| Feature | 87.6±0.1 ↑ | 95.5±0.2 ↓ | 81.7±0.2 ↓ | 36.0±0.3 ↓ | 94.3±0.6 ↑ | 25.3±0.7 ↓ | 4 |
| LAM | **88.1**±0.2 ↑ | **96.5**±0.3 ↑ | **86.0**±0.3 ↑ | **41.2**±0.2 ↑ | 93.5±1.8 ↑ | **29.8**±0.3 ↑ | **0** |

probability matching faithfully enforces the causal-invariant constraints on SS pairs, while logit and feature matchings enforce stronger conditions.

## 5.4 PERFORMANCES OF LAM

The results for LAM are shown at the bottom row of Table 1. On the first four datasets, it achieves the best OOD performance among all the five CR-based methods. As indicated by the arrows, it is the only CR-based method that consistently improves over ERM+DA on all the four datasets.

In LAM, a labelled SS pair $(x_k, \tilde{x}_k : y_k)$ is used only to regularize the contributions from feature units to the logit of the ground-truth class $y_k$. It does not impact other classes as should be. In the previous CR-based DG methods, on the other hand, the pair is used to regularize the entire feature, logit or probability vector for $x_k$. It affects other classes as well as $y_k$. This is a problem when a training example $x_k$ contains multiple objects of interest. Some objects that appear in the background of the main object in $x_k$ might be removed during data augmentation. In such a case, the features for those minor objects would be suppressed. To demonstrate the adverse consequences, we created a new version of the iWildCam dataset by adding a small image of another animal to the background of each image. The new dataset is named **iWildCam-N** (examples of this dataset are given in Appendix C). On this dataset, LAM still improves over ERM+DA. However, the performances of all the previous CR-based DG methods are substantially worse than that of ERM+DA.

In binary classification problems, there is essentially only one class (since the two classes are mutually exclusive). Here, the aforementioned advantage of LAM no longer exists. This explains why LAM is inferior to two previous methods on the binary classification dataset Camelyon. Nonetheless, it still significantly outperforms ERM+DA.

The results in Table 1 are based on targeted DA. In Table 8 (Appendix E), we provide results for iWildCam, iWildCam-N, and Camelyon using RandAugment (Cubuk et al., 2020), a more generic DA technique, to create augmented examples. Despite the change in DA, LAM consistently improves OOD performance and outperforms other CR-based DG methods.

## 5.5 COMPARISON OF LAM AND OTHER DG METHODS

Table 2 compares the OOD performances of LAM with those of the six representative other DG methods reviewed in Section 2. For the single-source methods RSC and SD, the augmented examples are simply added to the training set. For the multi-sources methods DANN, GDRO, IRM and V-REx, the augmented examples are treated as an additional training domain.

On the first four datasets, LAM outperforms all the six other DG methods. It beats them by large margins on iWildCam. As indicated by the arrows, the other DG methods are not as good as ERM+DA in the majority of the cases while LAM improves over ERM+DA on all the first four

Table 2: OOD performances of models trained using ERM, ERM+DA, LAM and other representative DG methods. Each model is trained three times and the standard deviation across runs is reported as ±. Arrows indicate changes relative to ERM+DA.

| | ImageNet-9 | NICO | PACS | iWildCam | Camelyon |
|---|---|---|---|---|---|
| ERM | 83.3±1.1 | 95.3±0.1 | 82.8±0.5 | 30.2±0.3 | 65.2±2.6 |
| ERM+DA | 86.0±1.0 | 95.9±0.3 | 84.5±0.5 | 36.5±0.4 | 90.5±0.9 |
| RSC | 86.4±0.2 ↑ | 94.0±1.8 ↓ | 84.3±0.6 ↓ | 32.7±0.9 ↓ | 91.6±0.3 ↑ |
| SD | 86.7±0.3 ↑ | 96.0±0.2 ↑ | 85.0±0.4 ↑ | 32.7±0.8 ↓ | **93.5±0.5** ↑ |
| DANN | 86.5±0.7 ↑ | 95.4±0.7 ↓ | 77.9±1.1 ↓ | 26.0±2.9 ↓ | 90.1±0.9 ↓ |
| GDRO | 83.7±0.8 ↓ | 91.8±1.2 ↓ | 83.5±0.5 ↓ | 37.0±1.0 ↑ | 92.2±0.9 ↑ |
| IRM | 87.1±0.2 ↑ | 93.5±0.2 ↓ | 83.2±0.4 ↓ | 31.7±0.1 ↓ | 90.8±2.6 ↑ |
| V-REx | 83.6±1.4 ↓ | 94.0±0.9 ↓ | 84.4±0.2 ↓ | 35.6±1.6 ↓ | 90.4±4.1 ↓ |
| LAM | **88.1±0.2** ↑ | **96.5±0.3** ↑ | **86.0±0.3** ↑ | **41.2±0.2** ↑ | **93.5±1.8** ↑ |

Table 3: The impact of the quality and quantity of SS pairs on LAM.

| Seg. | GrabCut | | | | | |
|---|---|---|---|---|---|---|
| # Pairs | 0% | 5% | 10% | 20% | 50% | 100% |
| Acc. | 83.3 | 88.1 | 88.5 | 88.6 | 89.7 | 90.4 |

| Seg. | Box | | Auto | | GrabCut | |
|---|---|---|---|---|---|---|
| | ERM+DA | LAM | ERM+DA | LAM | ERM+DA | LAM |
| Acc. | 85.2 | 85.9 | 83.9 | 86.6 | 86.0 | 88.1 |

datasets. On the binary classification dataset Camelyon, however, LAM is on par with SD method, and still outperforms ERM+DA.

## 5.6 EXTENDED EMPIRICAL ANALYSIS OF LAM

In our main experiments with ImageNet-9, only 5% of the training data was used to create SS pairs. To determine how the OOD performance of LAM is influenced by the number of SS pairs, we ran additional experiments with different amounts of SS pairs (in percentages of training data). The results are shown in Table 3 (left). It is clear that the increase in the quantity of SS pairs benefits and the availability of small proportion of SS pairs can significantly improve OOD performance already.

In our main experiments with ImageNet-9, the SS pairs were created with GrabCut (Rother et al., 2004). To determine how the OOD performance of LAM is influenced by the quality of SS pairs, we ran additional experiments where the SS pairs were created in less ideal ways, one with bounding boxes that come with ImageNet-9 (Box) and another using a semantic segmentation method (Long et al., 2015) (Auto). The results are shown in Table 3 (right). We see that LAM improves over ERM+DA in all cases. See Appendix B for more details and Appendix E for additional results.

We have argued that CR-based DG methods, particularly LAM, can help models focus more on core factors for prediction. To verify the claim, we examine the GradCAM (Selvaraju et al., 2017) saliency maps of the top predicted class by models trained using those methods. Examples are shown in Figure 9 and Figure 10. Those examples indicate that the CR-based DG methods are indeed effective in making model focus on core factors and LAM is the most effective among them.

## 6 CONCLUSION

In this paper, we have investigated the use of consistency regularization (CR) for domain generalization (DG). Theoretically, we argue that CR-based methods can boost the OOD performance of models because they make models more causal-invariant using semantic sharing (SS) pairs. Empirically, we find previous CR-based DG methods often improve over targeted data augmentation, but not consistently. We propose a novel CR-based DG method, LAM, that leverages class labels naturally associated with SS pairs. LAM improves over targeted data augmentation on all datasets tested in our experiments. The improvements increase with the quantity and quality of SS pairs. It outperforms previous CR-based methods on datasets with multiple classes, but may trail behind some of them on binary classification problems. A promising future direction is to develop more effective methods for creating SS pairs and apply LAM on them.

## REPRODUCIBILITY STATEMENT

The methodologies of both previous CR-based DG methods and our proposed CR-based DG method, Logit Attribution Matching (LAM), are thoroughly discussed in Section 4. Detailed information regarding the use of targeted data augmentation to create augmented examples for each dataset is provided in Section 5.1 and Appendix B .We implemented the targeted DA on iWildCam and Camelyon using the code provided by Gao et al. (2023).

Section 5.2 and Appendix D present the pre-trained models and other implementation specifics, including the selection of hyperparameters for various models. The implementation of other DG methods listed in Table 2 is based on the DomainBed (Gulrajani & Lopez-Paz, 2021).

All datasets used in our experiments are publicly accessible, except for iWildCam-N. The creation of the iWildCam-N are detailed in Section 5.4 and Appendix C. Upon paper acceptance, all related codes and the iWildCam-N dataset will be available online.

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

## APPENDICES

## A PROOFS

**Proof of Theorem 1**: Let us start with the ID cross entropy loss:

$$
\begin{aligned}
\ell_{P^s}(\hat{P}_\theta) &= \mathbb{E}_{(x,y)\sim P^s(X,Y)}[-\log \hat{P}_\theta(\hat{Y}=y|x)] \\
&= \mathbb{E}_{(x^c,x^n)\sim P^s(X^c,X^n), x\sim P^*(X|x^c,x^n), y\sim P^*(Y|x^c)}[-\log \hat{P}_\theta(\hat{Y}=y|x)] \\
&= \mathbb{E}_{(x^c,x^n)\sim P^s(X^c,X^n), x\sim P^*(X|x^c,x^n), y\sim P^*(Y|x^c)}[-\log Q_\theta(\hat{Y}=y|x^c)],
\end{aligned}
$$

where the last equality holds for some distribution $Q_\theta(Y|X^c)$ because $\hat{P}_\theta$ is causal-invariant. By removing irrelevant variables and re-arranging terms, we get:

$$
\ell_{P^s}(\hat{P}_\theta) = -\mathbb{E}_{x^c\sim P^s(X^c)}[\mathbb{E}_{y\sim P^*(Y|x^c)}[\log Q_\theta(\hat{Y}=y|x^c)].
$$

As the ID $\ell_{P^s}(\hat{P}_\theta)$ loss is minimized (over all model parameters and all model architectures), the inner expectation is maximized for any $x^c$ such that $P^s(x^c) > 0$.

Now, consider the OOD cross entropy loss of the target domain $P^t$ $\ell_{P^t}(\hat{P}_\theta)$. By symmetry, we have:

$$
\ell_{P^t}(\hat{P}_\theta) = -\mathbb{E}_{x^c\sim P^t(X^c)}[\mathbb{E}_{y\sim P^*(Y|x^c)}[\log Q_\theta(\hat{Y}=y|x^c)].
$$

We know from above that the inner expectation is maximized for all $x^c$ such that $P^s(x^c) > 0$. It is also maximized for any $x^c$ such that $P^t(x^c) > 0$, because

$$
\mathrm{supp}[P^t(X^c)] \subseteq \mathrm{supp}[P^s(X^c)].
$$

Therefore, the OOD loss $\ell_{P^t}(\hat{P}_\theta)$ is minimized. □

## B MORE DETAILS OF SS PAIR CREATION USING TARGETED DA

A SS pair is formed by a training example and an augmented example. The SS pair creation using targeted DA for each dataset has been introduced in Section 5.1. We provide more details and sample augmented examples here.

### B.1 IWILDCAM

For iWildCam, we utilized a targeted DA technique named Copy-Paste (same-y) from Gao et al. (2023). This DA method pastes the animal foreground onto a background image sampled from the same habitat where the same animal species has been observed. There is a category of images labeled "empty" in the iWildCam dataset. These images do not contain any animals and were used as background images when creating augmented examples. We used the segmentation for the animal foregrounds provided by Beery et al. (2021) to apply this DA. Sample augmented examples produced by this DA approach are provided in Figure 2.

### B.2 IMAGENET-9

In our main experiments, the synthetic images with a black background were used as augmented data for ImageNet-9. Those augmented examples were created based on the GrabCut segmentation. As also described in Section 5.6, to assess the performance of LAM under augmented examples in various qualities, we also considered the augmented examples created based on the bounding boxes and semantic segmentation. Specifically, we used the bounding boxes provided by the ImageNet (Deng et al., 2009) and semantic segmentation produced via FCN (Long et al., 2015), a semantic segmentation method. Sample augmented examples in various qualities are given in Figure 3.

| True label | Training example | Augmented example | True label | Training example | Augmented example |
|---|---|---|---|---|---|
| Giraffa Camelopa-rdalis | | | Loxodonta Africana | | |
| Aepyceros Melampus | | | Crax Rubra | | |

Figure 2: SS pairs created via Copy-Paste (same-y) DA for iWildCam. This DA method involves pasting the animal onto another image without animals sampled from the location where the same animal species has been observed.

| | Training example | Augmented example | | |
|---|---|---|---|---|
| **True label** | | **Bounding box** | **Semantic segmentation** | **GrabCut** |
| Dog | | | | |
| Wheeled vehicle | | | | |

Figure 3: Augmented examples in various quality created for ImageNet-9.

## B.3 NICO

For creating the augmented examples for NICO, we placed the foreground segmentation onto the background of a random image. We used GrabCut (Rother et al., 2004) to identify the foreground segmentation for 20 images in each class of NICO, which constituted about 5% of its training data. On average, the segmentation of an image took us around three seconds.

Since NICO does not have "empty" background images like iWildCam, we had to create synthetic background images. To do this, we removed the foreground in the image by coloring the image region corresponding to the foreground segmentation in black. We created the synthetic background images for all images with the foreground segmentation. When creating the augmented example, the foreground segmentation in the training example is pasted onto a randomly selected synthetic background image. See Figure 4 for some sample NICO augmented examples.

## B.4 CAMELYON

In dealing with the Camelyon dataset, we adopted the strategy outlined in Gao et al. (2023) to use the stain color jitter (Tellez et al., 2018) as the targeted DA to create the augmented examples. This technique transforms images by jittering their color in the hematoxylin and eosin staining color space. This DA addresses the style shift associated with the stain color resulting from diverse staining techniques used across different hospitals. It randomizes the average stain level in each image while maintaining all other information as predictive features. Sample augmented examples are shown in Figure 5.

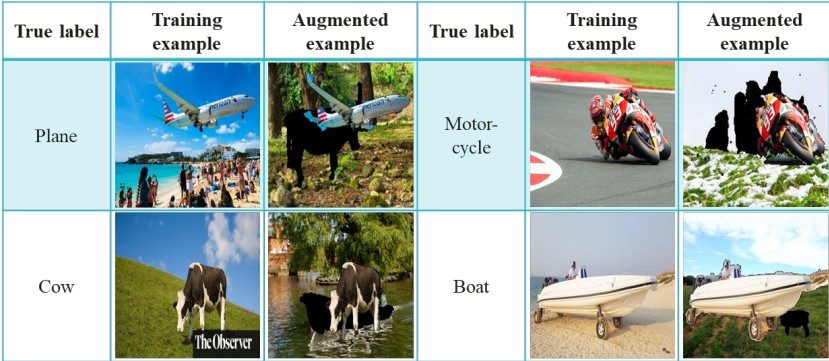

Figure 4: SS pairs created for NICO by placing the foreground segmentation onto a randomly selected synthetic background image.

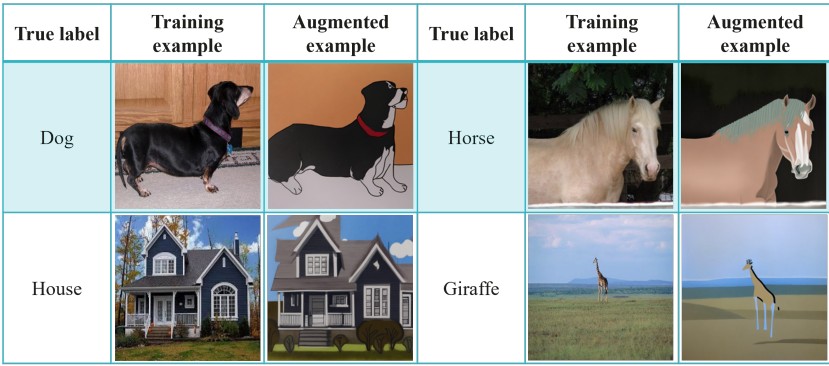

Figure 5: SS pairs created by stain color jitter for Camelyon dataset. This DA randomizes the average stain level in the image.

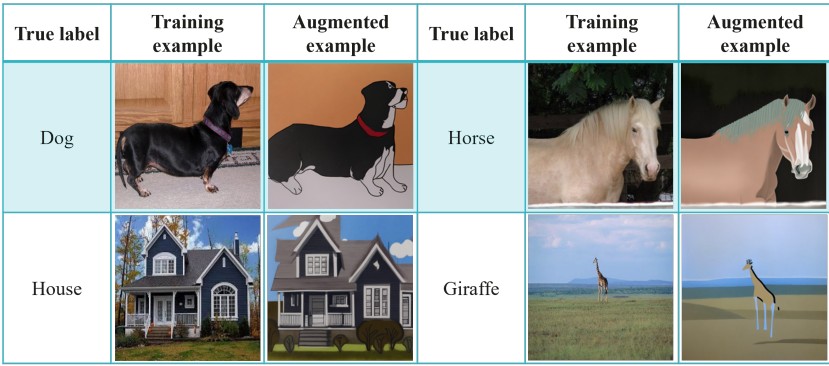

Figure 6: SS pairs created via StableDiffusion that generates augmented example from the training examples of the *photo* domain in PACS dataset. The prompt we use is "a minimalist drawing of a `class_name`, outline only, no texture" where `class_name` is the name of the true class label.

## B.5 PACS

To create SS pairs for PACS, we used StableDiffusion v2 (Rombach et al., 2022) to translate images from the *photo* domain of PACS into a different style. Given a training example $x_k$, we added a mild level of Gaussian noise to the latent representation of $x_k$, and then removed the noise under the guidance of a text prompt. The prompt we used is "a minimalist drawing of a `class_name`, outline

only, no texture" where class_name is the name of $y_k$. We chose this prompt because it produces the best visual quality among what we have explored. Finally, we decoded the generated noise-free latent representation, producing the corresponding augmented example $\tilde{x}_k$. See Figure 6 for some examples.

### B.6 Implementation summary of different datasets

We summarize setting of each dataset in model training, which includes how many training examples used to create augmented pairs and corresponding methods.

Table 4: Dataset details of shifts, pair quantity and methods to create aug. examples. We use "aug." as a shorthand for "augmentation".

| Dataset | Shift | Pair quantity. (% of training examples) | Method to create aug. examples |
|---------|-------|------------------------------------------|--------------------------------|
| ImageNet-9 | Background | 5% | Only preserve foreground objects, remove background as black |
| NICO | Background | 5% | Only preserve foreground objects, background replaced with one sampled from other images |
| iWildCam | Background | All images with animals | The animals are cut-and-paste to another image without animals taken at a different location where the same animals sometimes appear |
| iWildCam-N | Background | All images with animals | The animals are cut-and-paste to another image without animals taken at a different location where the same animals sometimes appear |
| PACS | Style | 100 samples for each class in Photo domain (P) | Employ StableDiffusion to transform the image style using the text prompt "a minimalist drawing of a class_name, outline only, no texture" |
| Camelyon | Style | 100% | Use augmentation of stain color jitter |

## C Details of iWildCam-N dataset

**iWildCam-N** dataset is an altered version of the iWildCam dataset, which includes extra background noise in addition to the original background shift in the iWildCam. This additional noise was created by inserting an animal foreground of a different animal species, sourced from a randomly selected image, onto the background of the image. To ensure the main semantic context of the image is not distorted due to the introduced noise, we limited the size of the introduced animal to be smaller than the pre-existing animal foreground and took steps to prevent overlap between the newly incorporated animal and the original animal foreground. We applied this operation on all images in the iWildCam dataset except for the images in the "empty" category, which do not contain any animals. The "empty" category was also excluded from the iWildCam-N dataset.

In Figure 7. We provide some examples of the iWildCam-N and their original images in the iWildCam to illustrate the background noise introduced in iWildCam-N.

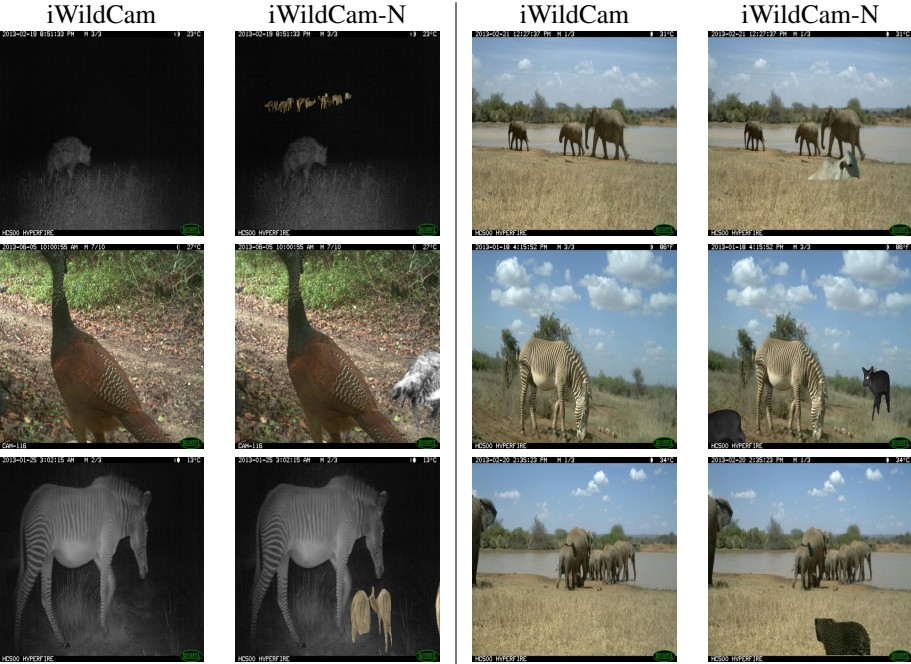

Figure 7: Sample images in iWildCam-N. The background noise is created by adding to the background of each image a small image of another animal.

# D    ADDITIONAL IMPLEMENTATION DETAILS

## D.1    IMPLEMENTATION DETAILS OF BASELINES

In Section 4.1, we introduce the formulation of previous CR-based methods. And in Section 2, we introduce other representative DG methods. According to their categories, the training data are organized differently to meet the assumption of each individual method. For datasets of ImageNet-9 and NICO, the setting is shown in Table 5. And for datasets of iWildCam, PACS and Camelyon, the setting is displayed in Table 6.

Table 5: Implementation details of LAM and the baselines on ImageNet-9 and NICO under background shift. The category of the methods is explained in section 5 of our paper. According to their categories, the training data are organized differently to meet the assumption of each individual method. On the other hand, the test data is the same for all methods: mixed-rand for ImageNet-9, and one unseen context per class for NICO. We use "aug." as a shorthand for "augmentation".

| Category | Method | Training data | Remark |
|---|---|---|---|
| Baseline | ERM | training examples | - |
| Data Aug. & Single-source | ERM-DA RSC SD | training examples + aug. examples† | As additional training data, aug. examples are mixed with training examples to train the model. |
| CR-based | LAM Prob. Match Logit Match Feature Match | training examples + aug. examples† | Only 5% training examples are paired with aug. examples, which are also as additional training data |
| Multi-source | DANN GDRO IRM VREx | domain #1: training examples domain #2: aug. examples | Training examples are regarded as one domain. Aug. examples form another. |

Table 6: Implementation details of LAM and the baselines on iWildCam, PACS and Camelyon. The category of the methods is explained in section 5 of our paper. According to their categories, the training data are organized differently to meet the assumption of each individual method. On the other hand, the test data is the same for all methods: unseen domains. We use "aug." as a shorthand for "augmentation".

| Category | Method | Training data | Remark |
|---|---|---|---|
| Baseline | ERM | training examples | - |
| Data Aug. & Single-source | ERM-DA
RSC
SD | training examples + aug. examples$^\dagger$ | As additional training data, aug. examples are mixed with training examples to train the model. |
| CR-based | LAM
Prob. Match
Logit Match
Feature Match | training examples + aug. examples$^\dagger$ | For PACS, only 100 training examples in Photo (P) domain are paired with aug. examples. For other datasets, each training example is paired with an aug. example. They are all also as additional training data. |
| Multi-source | DANN
GDRO
IRM
VREx | domain #1:
training examples-1
+aug. examples$^\dagger$-1
...
domain #m:
training examples-m
aug. examples$^\dagger$-m | aug. examples are put in the domain having corresponding training examples. |

## D.2 HYPERPARAMETER SETTING AND OTHER DETAILS

Each experiment is conducted on a single Nvidia V100 GPU. For datasets of ImageNet-9, NICO and PACS, we take LP-FT training scheme following (Kumar et al., 2022), while for other datasets we just take general fine-tuning. During LP, we take learning rate of 0.003 uniformly. And during FT, we take learning rate 3e-5 for ImageNet-9, NICO, PACS, and 3.49e-5 for iWildCam, 3.07e-3 for Camelyon. During LP, it trains models for 10 epochs. For FT, it trains 20 epochs for ImageNet-9, NICO, iWildCam, 40 epochs for PACS, and 10 epochs for Camelyon. The summary of hyperparameter setting is shown in Table 7.

Table 7: Hyperparameter setting for all the main experiments. SS-pair transformation refers to the transformation applied to training examples and corresponding augmented examples while training. If not specified otherwise, for other DG methods, the method-specific hyperparameters follow the default setting in DomainBed (Gulrajani & Lopez-Paz, 2021). For general training hyperparameters, if the ones are the same as CR-based methods, they will not be listed again. "bs" is the abbreviation of batchsize. For ImageNet-9, NICO, and PACS datasets, we take the linear probing (LP) and Fine-tuning (FT) scheme from (Kumar et al., 2022), while other datasets just take FT.

| Dataset | ImageNet-9&NICO | PACS | iWildCam | Camelyon |
|---|---|---|---|---|
| Model | CLIP ViT | CLIP ResNet-50 | ResNet-50 | DenseNet-121 |
| Pretrained | ImageNet Pretrained | | | False |
| Image Size | [224,224] | | [448,448] | [96,96] |
| LAM/ Logit Match/ KL | LP epochs: 10 FT epochs: 20 | LP epochs: 10 FT epochs: 40 | epochs: 20 | epochs: 10 |
| | LP learning rate: 0.003 FT learning rate: 3e-5 | | learning rate: 3.49e-5 | learning rate: 3.07e-3 |
| | LP training bs: 128 LP ss-pair bs: 256 FT training bs: 64 FT ss-pair bs: 64 | LP training bs: 48 LP ss-pair bs: 32 FT training bs: 48 FT ss-pair bs: 32 | training bs: 10 ss-pair bs: 10 | training bs: 128 ss-pair bs: 128 |
| | $\lambda = 10$    $\lambda = 0.5$ | $\lambda = 0.2$ | $\lambda = 5$ (LAM, KL) $\lambda = 0.05$ (Logit) | $\lambda = 10$ (LAM) $\lambda = 1$ (Logit, KL) |
| | ss-pair transform: RandCrop RandHorizontalFlip Normalize | ss-pair transform: RandCrop RandHorizontalFlip ColorJitter RandGrayscale Normalize | ss-pair transform: Normalize | ss-pair transform: Normalize |
| | N/A | $p = 0.9$ | N/A | |
| Feature Matching | $\lambda = 0.01$ | | $\lambda = 0.05$ | $\lambda = 0.1$ |
| JS | FT training bs: 32 FT ss-pair bs: 48 | FT training bs: 48 FT ss-pair bs: 48 | FT training bs:10 FT ss-pair bs: 20 | FT training bs: 128 FT ss-pair bs: 128 |
| Other Methods | LP training bs: 128 FT training bs: 64 | LP training bs: 48 FT training bs: 48 | training bs: 24 | training bs: 128 |

# E EXPERIMENTAL RESULTS WITH GENERIC-AUGMENTED EXAMPLES

For our main experiments reported in the Section 5, we used the targeted DA to create the augmented examples and evaluate the performance of CR-based DG methods under those augmented examples. Here we show some additional experiment results of those CR-based DG methods under the augmented examples produced by a generic DA method.

There are some popular generic DA methods such as Autoaugment (Cubuk et al., 2019), Fast Autoaugment (Lim et al., 2019), and RandAugment (Cubuk et al., 2020). They apply a sequence of PIL operations (such as rotation, shearing, autocontrast, RGB color jitter, etc.) onto images to generate a more diverse set of augmented examples. We select RandAugment Cubuk et al. (2020) as a representative of generic DA to generate the augmented examples for iWildCam, iWildCam-N and Camelyon, and then apply different CR-based DG methods based on those augmented examples. The results are reported in Table 8.

The results show that RandAugment can also improve model OOD performance on the three datasets, but the improvement is smaller compared to targeted DA. Similar to when augmented examples from targeted DA are used, when using augmented examples from RandAugment, the probability matching methods improve OOD performance on iWildCam over ERM+DA. However, the stronger conditions of logit matching and feature matching fail to make this improvement. LAM also produces significantly better performance than other CR methods when using RandAugment examples. For the iWildCam-N dataset, with RandAugment, only LAM can improve OOD performance over ERM+DA while other CR-based DG methods cannot, similar to when targeted data

augmentation is used. This again demonstrates the advantage of LAM in that it does not suppress features of other classes in the image.

The results on the Camelyon dataset again shows that LAM can effectively improve the model OOD performance even when the more generic data augmentation is used. The improvement of LAM over the ERM+DA baseline (which simply treats the augmented examples as additional training data) in fact larger when RandAugment is used (improve from 84.3 to 89.0) than when targeted DA is used (improve from 90.5 to 93.5).

Table 8: OOD performances of models trained using ERM, ERM+DA, previous CR-Based DG Methods, and LAM under different augmented data. The OOD performance of a model is assessed on held-out test domain(s) using Marcro F1 score of retrieval for iWildCam(-N) and accuracy for Camelyon. We do the following for models trained with the same augmented data: The best result is in **bold** and second best is underlined; Arrows indicate changes relative to ERM+DA.

|  |  | iWildCam | iWildCam-N | Camelyon |
|---|---|---|---|---|
| ERM | No DA | 30.2 | 15.3 | 62.5 |
| ERM+DA |  | 34.2 | 27.7 | 84.3 |
| Prob. (JS) |  | 34.7 ↑ | 26.4 ↓ | 83.4 ↓ |
| Prob. (KL) |  | 34.6 ↑ | 27.1 ↓ | 86.7 ↑ |
| Logit | RandAugment | 29.3 ↓ | 26.3 ↓ | 88.0 ↑ |
| Feature |  | 31.5 ↓ | 26.0 ↓ | 81.7 ↓ |
| LAM |  | **36.6** ↑ | **28.4** ↑ | **89.0** ↑ |
| ERM+DA |  | 36.5 | 28.2 | 90.5 |
| Prob. (JS) |  | 37.1 ↑ | 25.5 ↓ | **94.8** ↑ |
| Prob. (KL) |  | 40.3 ↑ | 26.3 ↓ | 92.8 ↑ |
| Logit | Targeted DA | 34.3 ↓ | 23.9 ↓ | 93.4 ↑ |
| Feature |  | 36.0 ↓ | 25.3 ↓ | 94.3 ↑ |
| LAM |  | **41.2** ↑ | **29.8** ↑ | 93.5 ↑ |

# F  WEIGHT ANALYSIS AND MORE VISUALIZATION RESULTS

## F.1  WEIGHT ANALYIS

We visualize the distribution of weight in model classifiers with Histograms in Figure 8. From it we can see that compared with probability matching, the weight distribution finetuned with LAM becomes sharper. It means that only a small proportion of weight units have large values while others are all near zero, which matches the target that only regularizes a subset of features.

## F.2  MORE VISUALIZATION RESULTS

We provide examples of saliency maps visualized by GradCAM. The examples in Figure 9 and Figure 10 are all randomly sampled from *mixed-rand* of ImageNet-9, which is our OOD test set for ImageNet-9.

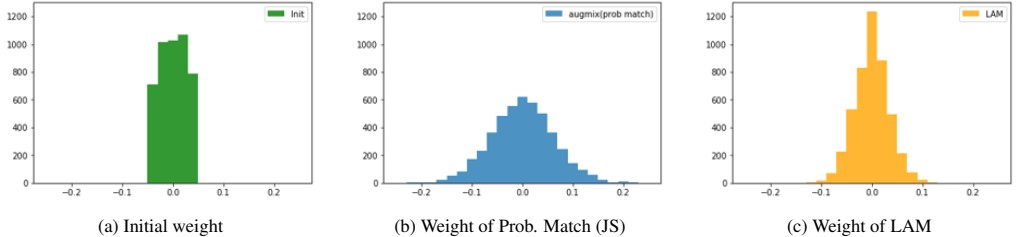

(a) Initial weight      (b) Weight of Prob. Match (JS)      (c) Weight of LAM

Figure 8: Distribution of weight in model classifiers with Histograms. The model is CLIP-pretrained ViT finetuned on ImageNet-9 dataset.

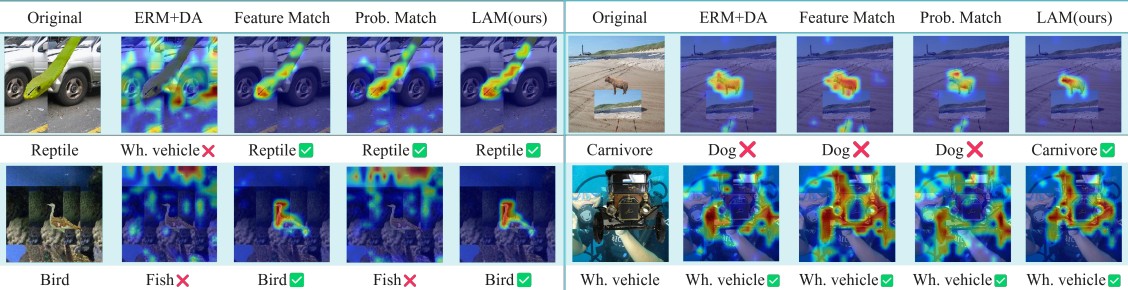

Figure 9: GradCAM saliency maps for the top predicted class by models trained using various methods. The examples are from the test domain of ImageNet-9.

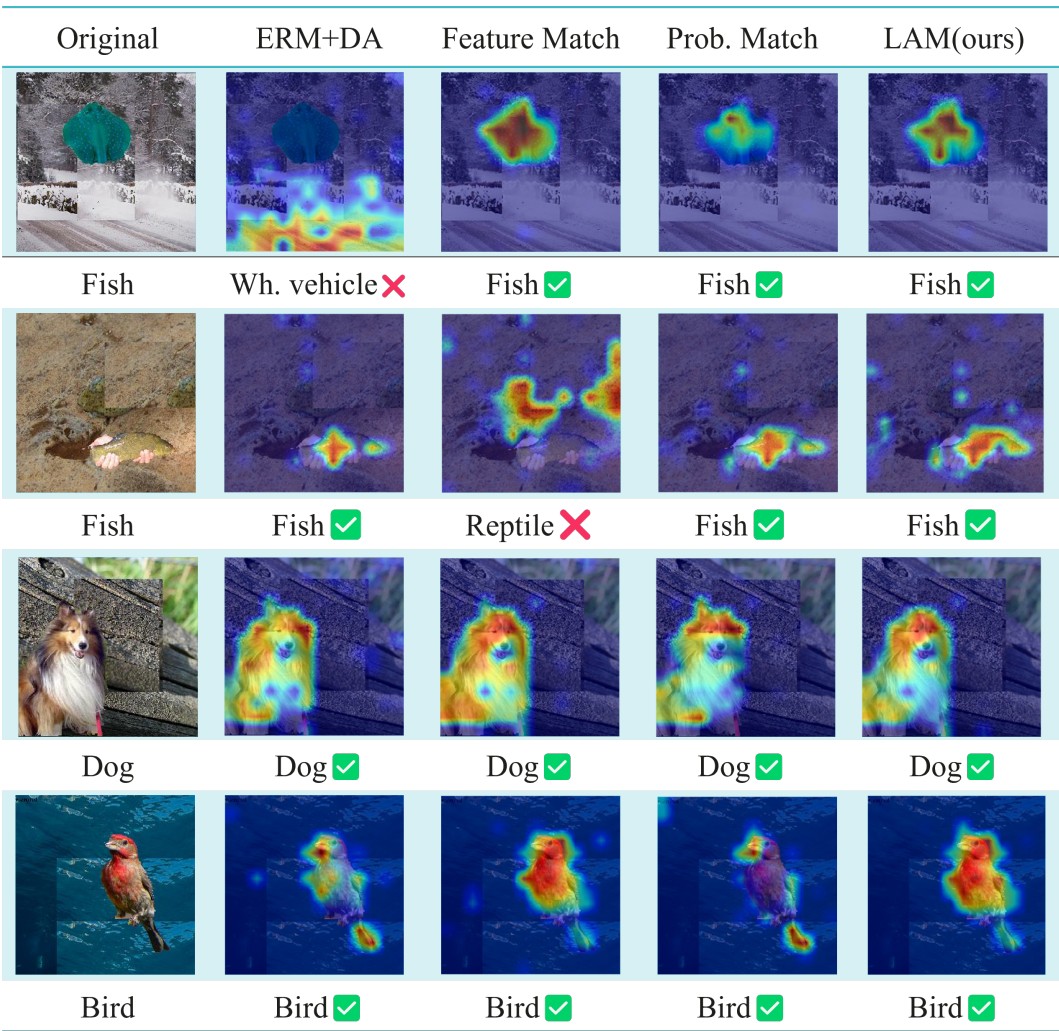

Figure 10: More GradCAM heatmaps of LAM compared with those of ERM-DA, Feature Matching and Probability Matching.

