# OpenReview forum: "Consistency Regularization for Domain Generalization with Logit Attribution Matching"
_ICLR.cc/2024/Conference — Submitted to ICLR 2024_

### Official Review · Reviewer_cViE · 2023-10-27

**Soundness:** 3 good
**Presentation:** 3 good
**Contribution:** 2 fair
**Rating:** 6
**Confidence:** 4

**Summary:**

This paper studies the magic of consistency regularization in domain generalization. First, authors claim that CR remains effective for DG and there are a few of existing approaches. Then, they study the theory behind the combination of CR and targeted augmentation. Finally, authors design their own approach called logit attribution matching to simply match the logits to further improve the performance. Experiments have shown its effectiveness.

------Post rebuttal

The response addressed my concerns and I increased the score to 6.

**Strengths:**

1. The paper presents a nice analysis of the consistency regularization in domain generalization, with interesting theoretical support.
2. Based on their theoretical analysis, the LAM approach is proposed which combines the existing targeted augmentation approach to further enhance the performance of CR-based DG.
3. Extensive experiments on several benchmark datasets have shown that the method brings improvements over ERM.

**Weaknesses:**

1. While I’m not an expert in causality, I hold doubt about the assumption in Figure 1, i.e., the data generation process. We certainly know that such generation process is an assumption, and other causality researchers can draw a completely different causal graph. Therefore, how can authors justify that this figure is practical and can be trusted? This is important since all analysis is based on this basic assumption.
2. I admire the effort to combine targeted augmentation with CR. But I do not think targeted augmentation is algorithmically novel since this is not general and hard to generalize to other datasets, given the wide popularity of DG in different applications. Therefore, the introduction of targeted augmentation is not efficient or general. This makes LAM deeply rely on the effectiveness of TA. I would like to know how can LAM be applied to other new domains where targeted augmentation is not realistic.
3. In the experiment section, I did not see any comparison with existing CR-based baselines, but only ERM variants. Did I miss anything?

**Questions:**

See the weakness. I'm extremely curious about the practical usage of LAM without the help from TA.

---

> ### Author Response · Authors · 2023-11-17
> **Response to the Reviews (1/2)**
>
> We thank the reviewer’s valuable time and constructive feedback. We are pleased to know that you found our paper “presents a nice analysis of the CR in DG, with interesting theoretical support”.
>
> Regarding the concerns and questions you've raised, we provide detailed explanations below.
>
> **1. “While I’m not an expert in causality, I hold doubt about the assumption in Figure 1, i.e., the data generation process. We certainly know that such generation process is an assumption, and other causality researchers can draw a completely different causal graph. Therefore, how can authors justify that this figure is practical and can be trusted? This is important since all analysis is based on this basic assumption.”**
>
> Our causal model assumes that the data generation process is as follows: (1) An example input $X$ is generated from two latent variables, $X^c$ and $X^n$. (2) These two variables, $X^c$ and $X^n$, are statistically correlated. (3) The label $Y$ is generated solely from $X^c$.
>
> Assumption (1) is a very common and general assumption which (at least) dates back to [Tenenbaum and Freeman (1996)](https://proceedings.neurips.cc/paper_files/paper/1996/file/70222949cc0db89ab32c9969754d4758-Paper.pdf). It basically says that data is generated from some latent variables (e.g., state of the physical world). As for Assumption (2), lifting it does not affect the results of our theory at all. It is kept simply to stress one of the main challenges of DG, that is, overcoming spurious correlations between $X$ and $Y$. The last assumption is also quite frequently made in the literature, together with the previous assumptions [1,2,3,4,5]. We can effortlessly point to many more examples in top conferences or journals [6,7,8,9].
>
> While it is sometimes more reasonable to posit a different connection between $Y$ and the other variables, the current assumption is nonetheless a very general one, encompassing a great number of real-world problems. For example, in SLAM problems, the label $Y$ is determined by the physical location of some agent in the world. As another example, in sentiment analysis, the label $Y$ is determined by the actual thought and feeling of the user.
>
> **2. “I would like to know how can LAM be applied to other new domains where targeted augmentation is not realistic.”**
>
> Common data augmentation (DA) methods based on basic operations such as resize, rotation, color jitter, and their random combination, RandAugment, all usually retain core factors and vary non-core factors for image classification tasks (e.g., the size of an object in an image could also be a non-core factor). After all, this is one of the most fundamental rules for data augmentation.
>
> In Table 8 (Appendix E), we compare LAM and other consistency regularization (CR) methods on iWildCam and iWildCam-N with semantic sharing pairs created by RandAugment. Similar to when targeted augmentation is used, LAM based on RandAugment also significantly outperforms other CR-based methods and ERM baselines in terms of OOD performance on the iWildCam dataset (OOD Macro F1-score – LAM: 36.6, the second best is Prob. (JS) at 34.7 and ERM baseline at 30.2).
>
> On the iWildCam-N dataset, with RandAugment, only LAM improves OOD performance over the ERM+DA baseline (which simply treats the augmented examples as additional training data), while other CR-based DG methods do not. This is in line with the results using targeted data augmentation and once again highlights LAM's superior ability of not suppressing features from different classes within the image.

---

> ### Author Response · Authors · 2023-11-17
> **Response to the Reviews (2/2)**
>
> **3. “I did not see any comparison with existing CR-based baselines, but only ERM variants. Did I miss anything?”**
>
> Yes, we had already made the comparisons in the paper. Most of the ERM variants (as you call them) in Table 1 are actually existing CR-based baselines. We mentioned this in the second paragraph of Section 4.1. Nevertheless, we would like to thank you for your question. We will make the connection between the variants and the baselines clearer in the paper.
>
> To be specific, *Prob. (JS)* and *Prob. (KL)* are essentially the CR methods proposed in  Representation Learning via Invariant Causal Mechanisms (ReLIC) [4] and AugMix [10], *Logit* refers to the regularization used in Conditional Variance Regularization (CoRE) [2], and *Feature* denotes that in MatchDG [3]. Those CR-based baselines differ in the way to apply regularization on semantic sharing pairs. The exact form of regularization for each of them is reviewed in Section 4.1. The findings from the comparisons are discussed in detail in Sections 5.3 and 5.4.
>
> ***
> Thank you for your reviews again. Please feel free to let us know if our responses well address your concerns, and feel free to let us know if you have any further questions or concerns. We would be more than pleased to engage in further discussion.
>
> **Reference:**
>
> [1] Liu, Chang, et al. "Learning causal semantic representation for out-of-distribution prediction." Advances in Neural Information Processing Systems 34 (2021): 6155-6170.
>
> [2] Heinze-Deml, Christina, and Nicolai Meinshausen. "Conditional variance penalties and domain shift robustness." Machine Learning 110.2 (2021): 303-348.
>
> [3] Mahajan, Divyat, Shruti Tople, and Amit Sharma. "Domain generalization using causal matching." International Conference on Machine Learning. PMLR, 2021.
>
> [4] Mitrovic, Jovana, et al. "Representation Learning via Invariant Causal Mechanisms." International Conference on Learning Representations. 2020.
>
> [5] Ye, Nanyang, et al. "Ood-bench: Quantifying and understanding two dimensions of out-of-distribution generalization." Proceedings of the IEEE/CVF Conference on Computer Vision and Pattern Recognition. 2022.
>
> [6] Lv, Fangrui, et al. "Causality inspired representation learning for domain generalization." Proceedings of the IEEE/CVF Conference on Computer Vision and Pattern Recognition. 2022.
>
> [7] Ouyang, Cheng, et al. "Causality-inspired single-source domain generalization for medical image segmentation." IEEE Transactions on Medical Imaging 42.4 (2022): 1095-1106.
>
> [8] Mao, Chengzhi, et al. "Causal transportability for visual recognition." Proceedings of the IEEE/CVF Conference on Computer Vision and Pattern Recognition. 2022.
>
> [9] Chen, Jin, et al. "Meta-causal Learning for Single Domain Generalization." Proceedings of the IEEE/CVF Conference on Computer Vision and Pattern Recognition. 2023.
>
> [10] Hendrycks, Dan, et al. "AugMix: A Simple Data Processing Method to Improve Robustness and Uncertainty." International Conference on Learning Representations. 2019.

---

> > ### Comment · Reviewer_cViE · 2023-11-20
> >
> > Thanks for the response. My concerns are mostly addressed. Thus, I would increase my rating.

---

> ### Author Response · Authors · 2023-11-22
> **Thanks for reading our response and increasing the score**
>
> Dear Reviewer cViE,
>
> We sincerely appreciate you for reading our response and increasing your score. We are always open and welcoming of any additional suggestions or comments you may have that could further enhance the quality of our paper. Please don't hesitate to share your thoughts, if any.
>
> Best Regards,
> The Authors

---

### Official Review · Reviewer_THPX · 2023-10-31

**Soundness:** 3 good
**Presentation:** 3 good
**Contribution:** 3 good
**Rating:** 6
**Confidence:** 3

**Summary:**

This paper presents a novel method called logit attribution matching (LAM) for improving domain generalization. Compared to existing consistency regularization methods (probability matching, logit matching, feature matching etc), the proposed method further adds class label information into the regularization term across semantic sharing pairs during data augmentation.

Experiments on a wide set of datasets show that LAM outperforms existing consistency regularization methods, and outperform domain adaptation approaches as well.

**Strengths:**

- The proposed idea is fairly simple by introducing an additional weight term over each feature unit and class label. Yet this simple weight term seems to be rather useful in improving the OOD performance over a wide set of image datasets.

- The authors did a fairly comprehensive set of experiments over 5 image datasets to demonstrate the superiority of the proposed method.

**Weaknesses:**

- Over the five datasets experimented, the augmentation is handpicked based on the characteristics of each dataset. This might make the disentanglement of causal/non-causal features relatively easier (i.e., the SS pairs better fit into this paper's motivation on $X^c$ and $X^n$). I wonder how the proposed approach works when the augmentation is agnostic to the datasets, i.e., what if you apply one of RandAugment / CutMix / AugMix to all the datasets as augmentation, and use LAM for regularization? How would the performance change?

**Questions:**

- Can the authors show how LAM works when the augmentation applied is dataset-agnostic? e.g., using augmentation methods like RandAugment / CutMix / AugMix for all the datasets?

---

> ### Author Response · Authors · 2023-11-17
> **Response to the Reviews**
>
> We thank the reviewer’s valuable time and constructive feedback. We are pleased to know that you found our proposed LAM is “fairly simple” and “useful in improving the OOD performance”.
>
> Regarding the question you've raised, we provide detailed explanations below.
>
> **“Can the authors show how LAM works when the augmentation applied is dataset-agnostic? e.g., using augmentation methods like RandAugment / CutMix / AugMix for all the datasets?”**
>
> Common DA methods based on basic operations such as resize, rotation, color jitter, and their random combination, RandAugment, all usually retain core factors and vary non-core factors for image classification tasks (e.g., the size of an object in an image could also be a non-core factor). After all, this is one of the most fundamental rules for data augmentation.
>
> In Table 8 (Appendix E), we compare LAM and other consistency regularization (CR) methods on iWildCam and iWildCam-N with semantic sharing pairs created by **RandAugment**.
>
> Similar to when targeted augmented examples are used, when utilizing RandAugment augmented examples, LAM also significantly outperforms other CR-based methods and ERM baselines in terms of OOD performance on the iWildCam dataset (OOD Macro F1-score – LAM: 36.6, the second best is Prob. (JS) at 34.7 and ERM baseline at 30.2).
>
> On the iWildCam-N dataset, with RandAugment, only LAM improves OOD performance over the ERM+DA baseline (which simply treats the augmented examples as additional training data), while other CR-based DG methods do not. This is in line with the results using targeted data augmentation and once again highlights LAM's superior ability of not suppressing features from different classes within the image.
>
> ***
>
> Thank you for your reviews again. Please feel free to let us know if our responses well address your concerns, and feel free to let us know if you have any further questions or concerns. We would be more than pleased to engage in further discussion.

---

> > ### Comment · Reviewer_THPX · 2023-11-22
> >
> > Thanks for the response. So it does seem like compared to targeted DA, LAM yields smaller improvements under dataset-agnostic augmentation (-4.6 on iWildCam and -1.4 on iWildCam-N). I think this additional experiment can be moved to the main text to show how the proposed method works in general (with more datasets like Table 1 would be ideal).

---

> ### Author Response · Authors · 2023-11-22
>
> Thank you for your response.
>
> As you suggested, we have applied various CR-based DG methods on the *CAMELYON17-WILDS* dataset using **RandAugment** to create augmented examples. The results are as below. For comparison, the results when **Targeted DA** is used are also copied below (from Table 1).
>
> |             Method            |      OOD Accuracy (RandAug)  |      OOD Accuracy (Targeted DA)
> |:-----------------------------:|:---------------:|:---------------:|
> |ERM|65.2$\pm$2.6 |65.2$\pm$2.6
> |ERM+DA| 84.3$\pm$2.2 |90.5$\pm$0.9
> |Prob. (JS)| 83.4$\pm$6.8|**94.8$\pm$1.2**
> |Prob .(KL)| 86.7$\pm$5.5|92.8$\pm$1.5
> |Logit| 88.0$\pm$1.4|93.4$\pm$0.3
> |Feature|81.7$\pm$ 5.2|94.3$\pm$ 0.6
> |LAM| **89.0$\pm$1.9**| 93.5$\pm$1.8
>
>
> The result again shows that LAM can effectively the model OOD performance even when the more generic data augmentation is used. The improvement of LAM over the ERM+DA baseline (which simply treats the augmented examples as additional training data) in fact larger when RandAugment is used (84.3-->89.0) than when targeted DA is used (90.5-->93.5).
>
> Just as you suggested, we will appropriately highlight the results with the generic data augmentation methods in the revised version of the paper.
>
> We appreciate your time in reviewing our paper. We hope the additional experiment results could enhance your confidence in our proposed LAM when used with the  dataset-agnostic augmentation methods.

---

### Official Review · Reviewer_H9LW · 2023-10-31

**Soundness:** 3 good
**Presentation:** 2 fair
**Contribution:** 2 fair
**Rating:** 3
**Confidence:** 4

**Summary:**

The paper studies the problem of domain generalization. It creates a theoretical model prescribing the relationship between the source and target domain, for which they argue the benefit of consistency regularization. The paper further presents a new consistency regularization scheme, referred to as Logit Attribution Matching (LAM). The key idea there is the match the logits of a pair of related examples while incorporating label information. Experimental study demonstrate performance improvements.

**Strengths:**

The main novelty of the paper are theoretical argument justifying the benefit of the consistency regularization and the proposal of the LAM, which take label information into account.  But to this reviewer, the novelty on both sides is thin. Theorem 1 holds nearly trivially; the LAM idea is also straight-forward.

**Weaknesses:**

1. Theorem 1 contains the very strong assumption that the target distribution of $X^c$ lies within the support of corresponding source distribution (Assumption 3) of the theorem. It is highly suspicious if in reality such a condition would hold true in conjunction with the first two assumptions. It seems that such a condition would only hold in a regime where transfer learning is easy.

2. In the description of LAM, it is not clear to me if the weights $\{w_{uy_k}\}$ are hyperparameters or if they are learned during training. If they are hyperparameters, how are they decided? and why not set them to 1 for each $(u, y_k)$?. If they are learned, what mechanism would force them to satisfy the two conditions listed on page 6 (lines 8 and 9)? Note that these weights and $f_\phi$ are learned together.

**Questions:**

See weakness.

---

> ### Author Response · Authors · 2023-11-17
> **Response to the Reviews**
>
> We thank the reviewer’s valuable time and constructive feedback.
>
> Regarding the concerns and questions you've raised, we provide detailed explanations below.
>
> **1. “Theorem 1 contains the very strong assumption that the target distribution of $X^c$ lies within the support of corresponding source distribution (Assumption 3) of the theorem. It is highly suspicious if in reality such a condition would hold true in conjunction with the first two assumptions.
> ”**
>
> We respectfully disagree that Assumption 3 of Theorem 1 is strong. On the contrary, we believe that this is one of the most fundamental assumptions required by DG. The reason is as follows:
>
> Imagine that a model had never seen certain $x^c$ during training, then it would not learn the corresponding $y$ of this $x^c$ unless we assume certain structures of $X^c$ that would allow us to infer the $y$ of the *unseen* $x^c$ from other *seen* $x^c$ (and their $y$).  However, such structural assumptions differ from task to task, and it is impossible to know the true underlying structure by only partially observing it without prior domain knowledge. Consequently, Assumption 3 is in fact a minimal assumption for optimal DG in general problems.
>
> **2. “In the description of LAM, it is not clear to me if the weights are hyperparameters or if they are learned during training. If they are hyperparameters, how are they decided? and why not set them to 1 for each (u, y_k)? If they are learned, what mechanism would force them to satisfy the two conditions listed on page 6 (lines 8 and 9)?”**
>
> The weights are learned during training. Moreover, they are *not* additional parameters introduced by LAM. They are just the weights of the classifier (assumed to be linear, so effectively the last layer) of the model. Consequently, the term $w_{uy_k}f_{\phi}^{u}(x_k)$  is exactly the attribution from a feature unit $u$ to the logit of certain class $y_k$ in the model. That is why we call it Logit Attribution Matching.
>
> To understand how the two conditions listed on page 6 are satisfied, note that the LAM regularization term is minimized if and only if either:
>
>  $w_{uy_k} = 0$ or  $f_{\phi}^{u}(x_k) = f_{\phi}^{u}(\tilde{x}_k)$,
>
> for every $u$ and $k$. This means that either the weight $|w_{uy_k}|$ is small or the feature difference $|f_{\phi}^{u}(x_k) - f_{\phi}^{u}(\tilde{x}_k)|$ is small (or both).
>
> In the final algorithm, the regularization term and the cross-entropy loss term are combined to establish the training objective as given in Equation (3).  This combination encourages the model to assign high weights to feature unit $u$ that are predictive and exhibit small feature difference $|f{\phi}^{u}(x_k) - f_{\phi}^{u}(\tilde{x}_k)|$. The core features are exactly such features. At the same time, the difference of features associated with higher weights would tend to be smaller than those associated with lower weights because of LAM. These are the driving forces behind the two conditions listed on page 6.
>
> ***
> Thank you for your reviews again. We also plan to appropriately include the points discussed above in the revised version of the paper.
>
> Please feel free to let us know if our responses well address your concerns, and feel free to let us know if you have any further questions or concerns. We would be more than pleased to engage in further discussion.

---

> > ### Comment · Reviewer_H9LW · 2023-11-20
> > **Your Point 1 is incorrect**
> >
> > I may have to re-read your paper to under your second point. But your point 1 is not correct.
> >
> > Cross-domain learnability also depends on the choice of hypothesis class, constructed (in part) from prior knowledge. The difference between source and target distributions on $X$ should be measured with respect to the hypothesis class. For example, a popular measure is ${\cal H}\Delta{\cal H}$ divergence as in Ben-David & Blitzer, "A theory of learning from different domains". It is possible that the target distribution is supported outside of the source distribution and yet the two distributions have small ${\cal H}\Delta{\cal H}$ divergence, providing learnability.

---

> ### Author Response · Authors · 2023-11-20
> **Further Clarification on Point 1 (1/2)**
>
> Thank you very much for following up on our response. Your point that cross-domain learnability also depends on the choice of hypothesis class is absolutely valid, but we fail to comprehend why this implies that our assumption 3 is strong.
>
> As you also mentioned, the choice of hypothesis class relies on prior knowledge of the task. This prior knowledge is essentially knowledge of the structures in data. If we do not know in advance how the $y$ of some unseen $x^c$ is related to other seen $x^c$ (and their $y$), then it is (almost) impossible to choose a hypothesis class that ensures optimal DG.
>
> While for some problems we may have sufficient prior knowledge and effective ways to shape the hypothesis space accordingly, for many other problems, it is not possible or at least very ineffective. In the latter problems, our assumption is indeed minimal (or close to being minimal), and our theorem conveys important theoretical insight about solving those general problems where we have little to no prior knowledge about the structures in data, or we do not know how to shape the hypothesis space as we want. That said, we do not claim that our assumption is weaker than that of Ben-David & Blitzer. In fact, it is neither weaker nor stronger from a theoretical perspective since the assumptions do not imply each other.
>
> In practice, our assumption can be approximately satisfied by covering the majority of potential cases of $x^c$ through the data collection process or simply limit the application to situations where we confidently know that the $x^c$ has been seen by the model during training (through calibration, out-of-distribution detection, etc.).

---

> > ### Comment · Reviewer_H9LW · 2023-11-21
> > **Consider this example**
> >
> > The task is sentiment classification. The source domain distribution is over the set of all movie reviews. The target domain  distributions is over all restaurant reviews.  This is a very typical setting of cross domain learning. Is the support of the target distribution contained in the support of source distribution? Obviously not.

---

> ### Author Response · Authors · 2023-11-20
> **Further Clarification on Point 1 (2/2)**
>
> Regarding the above matter in question of our theory, another equally important point we would like to make is that methods built upon the theory of Ben-David & Blitzer do not work well in practice for DG. One of the most prominent works in this direction is Domain-Adversarial Neural Network (DANN) [1] which aims to minimize ${\cal H}\Delta{\cal H}$ divergence by matching the feature distribution of different domains. The DG performance of DANN and example-level matching methods are shown below (the statistics are gathered from Table 1 and Table 2 of our paper).
>
> |                    | ImageNet-9 | NICO | PACS| iWildCam |Camelyon
> |:-----------------------------:|:-----------------------:|:---------------:|:---------------:|:---------------:|:---------------:|
> |ERM|$83.3$$\pm1.1$|$95.3$$\pm0.1$|$82.8$$\pm0.5$|$30.2$$\pm0.3$|$65.2$$\pm2.6$|
> |   **Example-level Matching**       | |       | | | |
> |Prob. (JS)|$86.0$$\pm0.4$|$95.0$$\pm0.3$|$84.3$$\pm0.3$|$37.1$$\pm0.4$|$94.8$$\pm1.2$ |
> |Prob. (KL)|$86.9$$\pm0.2$ | $95.4$$\pm0.2$  |$85.0$$\pm1.0$ |$40.3$$\pm0.3$ |$92.8$$\pm1.5$
> |Logit|$86.8$$\pm0.6$    |$95.3$$\pm0.2$   |$83.1$$\pm0.8$    |$34.3$$\pm0.5$   | $93.4$$\pm0.3$|
> |Feature|$87.6$$\pm0.1$   |$95.5$$\pm0.2$  | $81.7$$\pm0.2$ |$36.0$$\pm0.3$   | $94.3$$\pm0.6$  |
> |LAM| $88.1$$\pm0.2$ | $96.5$$\pm0.3$ |   $86.0$$\pm0.3$ |  $41.2$$\pm0.2$  | $93.5$$\pm1.8$|
> |   **Distribution-level Matching**       | |       | | | |
> |DANN|$86.5$$\pm0.7$  |$95.4$$\pm0.7$ |$77.9$$\pm1.1$ |$26.0$$\pm2.9$   |$90.1$$\pm0.9$ |
>
>
> Empirically, LAM and all other example-level matching methods significantly outperform DANN on PACS, iWildCam, and Camelyon. On ImageNet-9 and NICO, the performance of DANN is closer to that of example-level matching methods but still falls short of the best example-level matching methods. This stark contrast suggests that the assumptions made by Ben-David & Blitzer, especially the one on small ${\cal H}\Delta{\cal H}$ divergence between training and test domains, are probably much less realistic than ours under the setting of DG.
>
> Finally, we would like to stress that our theory offers a new perspective which in addition supports example-level matching, rather than only distribution-level matching. Example-level matching enables the utilization of additional information carried by paired examples as opposed to paired domains. The assumptions in our theory simply characterize the scope and limit of this alternative approach. Since whether some assumption is strong is a somewhat subjective matter, we have provided the above empirical evidence to support it. We hope this further clarification aids in better understanding the rationale behind our assumptions.
>
> ***
>
> Please feel free to let us know any further concerns or questions you might have regarding this point or any other. We will be glad to continue this discussion and provide further clarification if needed.
>
> [1] Ganin, Y., Ustinova, E., Ajakan, H., Germain, P., Larochelle, H., Laviolette, F., ... & Lempitsky, V. (2016). Domain-adversarial training of neural networks. The journal of machine learning research, 17(1), 2096-2030.

---

> ### Comment · Reviewer_H9LW · 2023-11-21
> **DANN does not minimize the correct divergence**
>
> DANN only aims at aligning the marginal distributions of the source and target on some representation space, with no intent to minimize the ${\cal H}\Delta {\cal H}$ divergence of Ben-David/Blitzer. The learning algorithm justified by Ben-David's theory is the paper by Zhang et al, "Bridging Theory and Algorithm for Domain Adaptation", ICML 2019. Your comment "this stark contrast suggests that the assumptions made by Ben-David & Blitzer, especially the one on small divergence between training and test domains, are probably much less realistic than ours under the setting of DG" based on comparing with DANN is ungrounded.

---

> ### Author Response · Authors · 2023-11-22
> **New Clarification on Point 1  (1/2)**
>
> Thank you for your engagement in the discussion. Since we have somewhat digressed from the main argument you raised in the review, we would like to first go back to your following statement.
>
>
> *1.  “Cross-domain learnability also depends on the choice of hypothesis class. It is possible that the target distribution is supported outside of the source distribution and yet the two distributions have small ${\cal H}\Delta{\cal H}$ divergence, providing learnability.”*
>
> As we mentioned, this statement is technically correct but at the same time vacuous for domain generalization (DG). **This is because DG does not have access to the target domain, unlike domain adaptation.** Without access to the target domain or any prior knowledge about data, there is no way to guarantee a small ${\cal H}\Delta{\cal H}$ divergence between the source and the target domain.
>
> In comparison, our theory does not rely on such assumptions about the hypothesis class, which are usually unreliable on its own due to the inaccessibility of the target domain in DG. Instead, we assume that $supp[P^t(X^c)] \subseteq supp[P^s(X^c)]$. This is our choice of assumption on the prior knowledge about data. Please note that this assumption is totally different from $supp[P^t(X)] \subseteq supp[P^s(X)]$. It is very common to have overlapping $P(X^c)$ but with non-overlapping $P(X)$ (e.g., PACS, iWildCAM).
>
> Theoretically, we prove a strong result, namely *optimal* DG, without making any requirement on the hypothesis class. It can be shown that $supp[P^t(X^c)] \subseteq supp[P^s(X^c)]$ is indeed necessary to prove the result (under the other two assumptions). From this theoretical perspective, our assumption is not that strong given the strength of the result. In addition, as we mentioned in our earlier response, there are many practical ways to (approximately) satisfy this assumption.
>
> Empirically, we have shown that the derived algorithms of our theory work well on multiple representative DG problems. Here, we would like to emphasize that we do not claim that our assumption applies to every DG problem, nor does it have to be strictly satisfied for the algorithms to work reasonably well. Meanwhile, there are many typical and practical DG problems that do satisfy our assumption to a large degree. For instance, object recognition across different backgrounds/styles, disease prediction across different population groups, speech recognition across different noise levels, face recognition across different lighting conditions and so on. From this empirical perspective, our assumption should not be regarded as “strong” either.
>
> ***
>
> We next respond to your more recent comments.
>
> *2. “The task is sentiment classification. … Is the support of the target distribution contained in the support of source distribution? Obviously not.”*
>
> Recall that our assumption is $supp[P^t(X^c)] \subseteq supp[P^s(X^c)]$ instead of $supp[P^t(X)] \subseteq supp[P^s(X)]$, where $X^c$ refers to the core features. It is disputable (rather than obvious) whether the assumption, $supp[P^t(X^c)] \subseteq supp[P^s(X^c)]$, holds in the example you give. Basically, $X^c$ captures the sentiment of the users. Sentiment can be very complicated, comprising of multiple basic emotions such as joy, surprise, sadness, fear, anger, etc. These basic emotions are to a large degree shared across domains, regardless of whether it is a sentiment about a movie or a restaurant. Meanwhile, the label $Y$ is just a coarse summarization of the sentiment. In this sense, the example you give actually satisfies our assumption.

---

> ### Author Response · Authors · 2023-11-22
> **New Clarification on Point 1 (2/2)**
>
> *3. “DANN only aims at aligning the marginal distributions of the source and target on some representation space, with no intent to minimize the ${\cal H}\Delta {\cal H}$ divergence of Ben-David/Blitzer. The learning algorithm justified by…”*
>
> Thanks for letting us know the MDD method proposed in "Bridging Theory and Algorithm for Domain Adaptation" *ICML* (2019). Regarding the methods built upon the theory of Ben-David & Blitzer, including DANN and MDD (note that DANN is indeed related to the minimization of distribution divergence, while will be discussed at the end of this post), we would like to make further remarks on them.
>
> Again, in this paper, we focus on domain generalization (DG) not domain adaptation. While DANN and MDD aim to reduce distribution divergence between source and known target domains, they are not directly applicable to DG where target domains are unseen. In the DG context, we can only minimize divergence among known source domains. Hence, in our experiments, DANN is treated as a multi-source DG method, where we reduce divergence between the original image domain and the augmented image domain. We are also training the MDD method on the Camelyon dataset to minimize distribution divergence of the original domain and augmented domain now. We will get back to you once we get the results.
>
> It is true that, as you said, DANN aims at aligning the marginal distributions of the source and target on some representation space. This implies that DANN is at least related to the minimization of distribution divergence. As demonstrated in Theorem 2 and discussed in Section 3.3 of the [the original paper](https://arxiv.org/pdf/1505.07818.pdf) of DANN, the distribution-aligning objective in DANN is an upper bound for the ${\cal H}$ divergence of Ben-David/Blitzer proposed in "Analysis of representations for domain adaptation." *NIPS* (2006). Whether it is also an upper bound for the ${\cal H}\Delta {\cal H}$ divergence is not discussed in the paper. In theory, when the marginal distributions are perfectly aligned by DANN, both ${\cal H}$ divergence and ${\cal H}\Delta {\cal H}$ divergence are minimized.

---

> ### Author Response · Authors · 2023-11-23
> **Results of MDD method on the Camelyon dataset**
>
> We trained the MDD method on Camelyon dataset to minimize distribution divergence of original domain and augmented domain. We strictly followed the experiment settings in the released code (https://github.com/thuml/MDD). Despite extensive exploration of different hyperparameters, the OOD accuracy of the best model trained by MDD still significantly underperformed when compared to example-matching methods, even falling short of the performance of DANN.
>
> The table below illustrates the OOD accuracy of models trained with MDD under varying margin factors ($\gamma$) and learning rates. In the paper experiments, $\gamma$ was chosen from the set {2, 3, 4}. Here, we have expanded the exploration to include {1, 2, 3, 4, 5, 6}.
>
> |        learning rate            | margin factor $\gamma$  | OOD accuracy
> |:-----------------------------:|:-----------------------:|:---------------:|
> |0.003 | 1 | 71.86|
> |0.01 | 1 |  73.27|
> |0.003 | 2 |66.78|
> |0.01 | 2 |66.38 |
> |0.003 | 3 |**83.14**|
> |0.01 | 3 | 61.96|
> |0.003 | 4 |61.77|
> |0.01 | 4 | 70.26|
> |0.003 | 5 |76.38|
> |0.01 | 5 | 63.34|
> |0.003 | 6 |64.81|
> |0.01 | 6 | 49.55 |
>
> The highest OOD accuracy achieved was 83.14%, in contrast to the 94.8% of the best example-matching methods. The performance of each example-matching method is detailed in the table in our prior response "Further Clarification on Point 1 (2/2)". The results again show that, in the setting of domain generalization, the unjustified distribution-level matching methods work not better than the example-level matching methods which are well supported by our theorem.

---

### Official Review · Reviewer_9AoB · 2023-10-31

**Soundness:** 2 fair
**Presentation:** 1 poor
**Contribution:** 2 fair
**Rating:** 3
**Confidence:** 3

**Summary:**

Models can be trained to generalize to new domains (called domain generalization) through augmenting training data. Another approach to domain generalization is through consistency regularization, which enforces that the model should make similar predictions on similar inputs.  This paper proposes using consistency regularization on top of data augmentation; that is, enforcing similarities (at the logit, output, embedding level) on pairs of unaugmented and augmented samples. The paper provides a theoretical result stating that a model that is causal invariant (E.g. $\hat{P}(Y | x) = \hat{P}(Y | \tilde{x})$) and minimizes loss in-distribution will also minimize out-of-distribution loss. The paper finally proposes a new form of consistency regularization, called logit attribution matching (LAM), which encourages feature matching on features that are strongly associated with the true label; this is more granular than previous approaches. The performance of consistency regularization on top of targeted data augmentation is compared to using standard data augmentation to expand the training dataset, and LAM is compared against other consistency regularization methods.

**Strengths:**

Quality:
- Theoretical results motivate why causal invariant property is important for domain generalization.
- LAM outperforms both DG methods and CR methods

Clarity:
- Toy illustration in figure 1 made theoretical result and setup more clear.

Significance:
- Handling OOD settings is an important problem in machine learning.

**Weaknesses:**

Quality:
- Theory is not connected to LAM. Since the condition is $\hat{P}(y | x) = \hat{P}(y | \tilde{x})$, why does probability matching not work well? Why does LAM work better?
- The theoretical model's connection to data augmentation is also rather weak. Can you show that your choice of data augmentation is retaining $x^c$ and changing $x^n$?

Originality:
- Having trouble understanding why CR on top of DA is a contribution. In the related work you say that CR can use different data augmentation strategies as well as alternate ways to pair up samples.
- This paper combines two well-studied ideas into one with a new theoretical result and a new consistency regularization term, but the theory and the new term could be more motivated.

Clarity:
- Minor nit: this paper has many abbreviations (DG, DA, OOD, CR, CLD). I found that use of DG and DA were a bit confusing the first time I read the paper, and would prefer having more sentences with the full words, at least in the introduction.
- Unclear how contributions in the introduction are related. It reads like a list of ways to improve performance but don't feel well-motivated.

**Questions:**

- Theory is not connected to LAM. Since the condition is $\hat{P}(y | x) = \hat{P}(y | \tilde{x})$, why does probability matching not work well? Why does LAM work better?
- The theoretical model's connection to data augmentation is also rather weak. Can you show that your choice of data augmentation is retaining $x^c$ and changing $x^n$?

---

> ### Author Response · Authors · 2023-11-16
> **Response to the Reviews (1/3)**
>
> We thank the reviewer’s valuable time and constructive feedback. We are pleased to know that you found our “theoretical results” and “toy illustration in figure 1” are motivating.
>
> Regarding the concerns and questions you've raised, we provide detailed explanations below.
>
> **1. "Theory is not connected to LAM. Why does probability matching not work well? Why does LAM work better?"**
>
> Thank you very much for pointing this out. As you kindly suggested, we provide below a clear and concise explanation of how the logit attribution matching (LAM) is connected to the theory and why LAM works better than probability matching.
>
> In LAM, the labelled pair $(x_k, \tilde{x}_k: y_k)$ is used whereas the unlabelled pair $(x_k, \tilde{x}_k)$ is used in prior CR-based methods. The key here is to realize that the pairing of $x_k$ and $\tilde{x}_k $ in the labelled pair $(x_k, \tilde{x}_k: y_k)$ is *less informative* than the corresponding unlabelled pair $(x_k, \tilde{x}_k)$. In fact, $(x_k, \tilde{x}_k)$ means that $x_k$ and $\tilde{x}_k$ contain the same semantic information about ALL classes, while $(x_k, \tilde{x}_k: y_k)$ means that $x_k$ and $\tilde{x}_k$ contain the same semantic information about ONE PARTICULAR class $y_k$.
>
> This distinction between labelled and unlabelled pairs matters because, in practice, it is much harder to ensure that $x_k$ and $\tilde{x}_k$ hold identical information about all classes of interest than a single class $y_k$, especially when the total number of classes is large. For instance, images labelled with class $y_k$ may include small objects of other classes $y_k'$ in the background. This kind of label noise is pervasive in machine learning datasets [1]. When creating the augmented example $\tilde{x}_k$ from $x_k$, it is much easier to retain information about only $y_k$ since we do not need to care about features of other classes when augmenting the data to randomize non-core features $x^n_k$.
>
> In the aforementioned realistic scenarios, matching the probability distributions over ALL classes is not justified and would lead to sub-optimal solutions (the same can be said about matching the entire feature vectors, $f(x_k)$ and $f(\tilde{x}_k)$). Instead, the logical thing to do is to match only the features that are important for the $y_k$ class. Those considerations lead to LAM. In LAM, regularization is applied only to the labelled pairs $(x_k, \tilde{x}_k: y_k)$, with the assumption that $x_k$ and $\tilde{x}_k$ share the same information of $y_k$. As explained earlier, this assumption is much more practically reasonable since it significantly lowers the requirement on the quality of SS pairs. The advantage of LAM in this respect is specifically demonstrated through the iWildCam-N dataset with amplified label noise. As shown in Table 1 and discussed in Section 5.4 of the paper, on this dataset, only LAM can improve OOD performance over the ERM+DA baseline (which simply use the augmented examples as additional training data), while all previous CR-based methods that regularize across the entire feature, logit, or probability vector fall short of the baseline.
>
> To summarize, the theory essentially provides us with a general understanding of optimal DG under ideal conditions where probability matching across all classes works best. However, when practical aspects of the problem are also considered, we must relax the condition. Under the relaxed condition, LAM is the best choice that we currently have in the literature (to the best of our knowledge).

---

> ### Author Response · Authors · 2023-11-16
> **Response to the Reviews (2/3)**
>
> **2. "The theoretical model's connection to data augmentation is also rather weak. Can you show that your choice of data augmentation is retaining $x^c$ and changing $x^n$?"**
>
> Targeted augmentation incorporates human knowledge to control and vary non-core factors while retaining core factors. Take the iWildCam dataset as an example. Since different animals usually live in different habitats, the non-core factors that the models tend to exploit are the backgrounds in the images. So, for this dataset, we can retain $x^c$ and change $x^n$ by copying the animals from one image to another image taken at a different location where the animals might appear. This choice of data augmentation effectively changes spurious background factors while preserving robustly predictive animal traits in the training examples [2]. This fact is reflected by the results in Table 1, showing that by simply adding these augmented examples to the training data, the learned model already significantly outperforms its counterpart trained without the augmented examples (OOD Macro F1-score: 36.5 vs. 30.2). In addition to that, LAM further improves the performance to 41.2. Similar results are also observed on various datasets. Such significant improvements cannot be explained if $x^c$ is not retained or $x^n$ is not changed by the chosen augmentation.
>
> Besides, common DA methods based on basic operations such as resize, rotation, color jitter, and their random combination, RandAugment, all usually retain core factors and vary non-core factors for image classification tasks (e.g., the size of an object in an image could also be a non-core factor). After all, this is one of the most fundamental rules for data augmentation. In Table 8 (Appendix E), we compare LAM and other consistency regularization (CR) methods on top of RandAugment. It shows that CR on top of RandAugment also significantly outperforms the ERM baseline on the iWildCam dataset.
>
> **3. "Having trouble understanding why CR on top of DA is a contribution? In the related work you say that CR can use different data augmentation strategies as well as alternate ways to pair up samples."**
>
> We would like to take this great opportunity to clarify the first contribution of our paper. By “empirically showing that CR, if done properly, can improve OOD performance when combined with DA”, we do not claim that we are the first to propose CR methods for DG. Instead, we were referring to the fact that there had not been an extensive discussion and comparison of existing CR methods for DG until our work. We believe this is **the first time** that multiple methods are systematically evaluated together under the same evaluation protocol and data. By benchmarking existing CR methods and showing their effectiveness (as well as limitations), we hope that our work draws more attention from the research community to this promising but relatively unexplored direction. We will carefully revise the introduction of our paper to better reflect the above points.
>
>
> **4. "I found that use of DG and DA were a bit confusing the first time I read the paper, and would prefer having more sentences with the full words, at least in the introduction."**
>
> Thanks for your kind advice. You are absolutely right. We will pay attention to it and address it in the revised version of the paper.

---

> ### Author Response · Authors · 2023-11-16
> **Response to the Reviews (3/3)**
>
> **5. "Unclear how contributions in the introduction are related. It reads like a list of ways to improve performance but don't feel well-motivated."**
>
> The starting point of our paper is that data augmentation (DA) is quite important and effective for DG. A recent study shows that targeted augmentations can significantly improve OOD performance in various real-world datasets [2]. Furthermore, DA creates abundant semantic sharing pairs which can be naturally utilized in consistency regularization (CR). To better understand and harness CR for DG, we first theoretically show that CR on top of DA can lead to optimal DG and then systematically evaluate existing CR approaches on a scale and rigor that has not seen before. In addition, we propose a new CR method, LAM, to achieve even better performance compared to previous CR-based DG methods and other representative DG methods. Finally, the connection between LAM and our theory is discussed in the answer to the Question 1 above. If you find the added discussions are appropriate and helpful, we will revise the introduction section of the paper to reflect this connection better.
>
> ***
> Thank you for your review again. Please feel free to let us know if our responses well address your concerns, and feel free to let us know if you have any further questions or concerns. We would be more than pleased to engage in further discussion.
>
> [1] Northcutt, Curtis G., Anish Athalye, and Jonas Mueller. "Pervasive Label Errors in Test Sets Destabilize Machine Learning Benchmarks." In Thirty-fifth Conference on Neural Information Processing Systems Datasets and Benchmarks Track. 2021.
>
> [2] Irena Gao, Shiori Sagawa, Pang Wei Koh, Tatsunori Hashimoto, Percy Liang; Proceedings of the 40th International Conference on Machine Learning, PMLR 202:10800-10834.

---

> > ### Comment · Reviewer_9AoB · 2023-11-22
> >
> > Thank you for your response.
> > For my first point on theory and LAM, I do not quite understand your point. I get the motivation of LAM and why probability matching does not work as well in cases, but I do not understand the exact "practical aspects of the problem" that that are different from Theorem 1 that implies that the causal invariant condition from Theorem 1 should be relaxed. In particular,
> >
> > > Among the aforementioned CR-based DG methods, probability matching aims to directly enforce
> > the equality (2) on the SS pairs. This makes the prediction model more causal-invariant, which
> > is conducive to good OOD performance according to Theorem 1.
> >
> > If probability matching is good according to Theorem 1, what differences between Theorem 1 and practice motivate LAM? Thank you in advance and apologies for my confusion.

---

> > > ### Author Response · Authors · 2023-11-22
> > >
> > > Thank you very much for following up on our response.
> > >
> > > For your question, consider the second pair of images on the first row of Figure 3 (Appendix B), and imagine a setting where we are interested in only two classes, *elephant* and *lake*. Denote the pair as $(x, x’)$.  The causal-invariant condition of Theorem 1 requires both $P(lake|x) = P(lake|x’)$ and $P(elephant|x) = P(elephant|x’)$. The first part clearly does not hold.
> > >
> > > In fact, $x’$ was created from $x$ to preserve information for the *elephant* class only. In this sense we have a labelled SS pair $(x, x’: elelphant)$.   Applying probability/feature matching on labelled pairs is clearly not justified. Hence, LAM is proposed.  In our response above, we argue that labelled SS pairs are common in practice.
> > >
> > > Empirical results reported in Table 8 (Appendix E) indicate that it is beneficial to regard pairs created using RandAugment as labelled. Since RandAugment does not require information on the dataset,  the SS pair created by it is unlabelled. This means, in this case, among all CR-based DG methods, only LAM uses label information while all others do not.
> > >
> > > ***
> > > Please feel free to let us know any further concerns or questions you might have regarding this point or any other. We will be glad to continue this discussion and provide further clarification if needed.

---

### Official Review · Reviewer_4UiZ · 2023-11-02

**Soundness:** 3 good
**Presentation:** 3 good
**Contribution:** 2 fair
**Rating:** 5
**Confidence:** 4

**Summary:**

This paper aims to enhance the consistency regularization method for domain generalization by incorporating a logit attribution matching approach.
The authors first revisit the domain generalization (DG) problem through the causal latent decomposition (CLD) model. This model indicates that DG adheres to the concept of causal-invariant prediction, wherein the predicted labels for a semantic sharing pair remain consistent with diverse non-core factors, as long as the core factors remain unchanged. They then introduce a theorem of Conditions for Optimal DG and unveil consistency regularization as a potential optimal solution for DG, subject to certain assumptions. Existing methods, such as probability matching, logit matching, and feature matching, can all be treated as special cases of this optimal solution. The authors then develop the Logit Attribution Matching (LAM) regularizer, building upon the feature matching method. This approach introduces weights on each dimension of the features, corresponding to each label y. It is hoped that this design allows the model to pay more attention to core factors than non-core factors and improve OOD performance.

**Strengths:**

1. The writing of this paper is clear and the idea is easy to follow.
2. It is interesting to revisit domain generalization from a causal latent decomposition perspective and highlight the core and non-core factors that are not considered in previous works.
3. It is also interesting to utilize the Optimal DG theorem to summarize the existing consistency regularization methods into a general framework.
4. A thorough experiment is conducted in the main paper content as well as the appendix to evaluate and analyze the proposed methods.

**Weaknesses:**

1. The contributions may not be very significant because consistency regularization for DG has already been extensively studied in previous research and the proposed method only makes simple modifications to the existing feature matching method. Several concepts in this paper are borrowed from previous ideas, such as targeted augmentation and causal-invariant prediction.

2. The theorem does not serve as a supporting foundation for the proposed logit attribution matching method. It seems that the theorem and the techniques are divided and tell two different stories. Concretely, the theorem just reveals that consistency regularization can be an optimal solution for DG. It is not connected to why logit attribution matching is needed to deal with core and non-core factors. Before this paper can be accepted, it is necessary that it undergoes revision to ensure a clear and concise explanation of how the theorem serves as a driving force behind the techniques employed.

**Questions:**

Is it possible to build logit attribution matching upon probability matching or logit matching models?
Are they better or worse than building upon a feature matching model?
Can the authors provide more analysis and explanations of why you just choosing feature matching methods?

---

> ### Author Response · Authors · 2023-11-16
> **Response to the Reviews (1/2)**
>
> We thank the reviewer’s valuable time and constructive feedback. We are pleased to know that you found our work to be "clear" and "easy to follow". Your appreciation for our "thorough experiments" and our theorical analysis to revisiting domain generalization from a "causal latent decomposition perspective" is encouraging.
>
> Regarding the concerns and questions you have raised, we provide detailed explanations below.
>
> **1. "The contributions may not be very significant. Consistency regularization for DG has already been extensively studied. Several concepts in this paper are borrowed from previous ideas, such as targeted augmentation and causal-invariant prediction."**
>
> - We greatly appreciate the chance to clarify the contributions of our paper; however, we respectfully disagree that consistency regularization (CR) for DG has been extensively studied. Considering the sheer volume of DG literature, CR for DG is relatively unexplored. In fact, it is not even considered as an established category of DG methods in recent DG survey papers (see Table 3 of [1] and Figure 3 of [2]).
>
>   In addition, there has not been an extensive discussion and comparison of existing CR methods for DG until our work. We believe this is **the first time** that multiple methods are systematically evaluated together under the same evaluation protocol and data. The results under targeted augmentation and a more generic form of data augmentation (RandAugment) are presented in Table 1 and Table 8 respectively. Discussion of the results is also made in Section 5.3 and Appendix E. By benchmarking existing CR methods and showcasing the effectiveness of CR when done properly for DG, we hope that it draws more attention from the research community to this promising direction.
>
> - Almost every paper borrows some concepts from previous ideas. We genuinely do not think this is an originality issue of our paper as what we propose is quite different from targeted augmentation [3] and invariant causal prediction (ICP) [4].
>
>   To be specific, our causal DG model differs fundamentally from ICP: in our model, $X^c$ is unobserved, unlike ICP which assumes that $X^c$ is observed and disentangled from $X^n$. This is a very important difference as our model is thereby compatible with unstructured data modalities like vision and text. With this more practical assumption, our causal DG model and theorem provide more general insights on the performance gains of consistency regularization (CR) for DG. Besides, the focus of this work is the generalization effect brought by CR between original and augmented data. Targeted augmentation is just one of the possible ways to create augmented examples. Other data augmentation methods also work to certain extents (e.g., RandAugment in Table 8).
>
> - Our final contribution is that we introduced a new CR approach: Logit Attribution Matching (LAM), following the theoretical discussion and empirical evaluation of prior CR-based DG methods. Empirically, LAM surpasses other CR methods and representative DG methods on multiple real-world datasets as shown in Table 1 and 2. In response to your third question below, we will delve deeper into the connection between LAM and our theorem and explain why it provides superior performance compared to previous CR-based DG methods.
>
> **2. "Is it possible to build logit attribution matching upon probability matching or logit matching models? Are they better or worse than building upon a feature matching model? Can the authors provide more analysis and explanations of why you just choosing feature matching methods?"**
>
> Thank you for your questions, but we suspect that there might be some misunderstanding. To facilitate better understanding, we will make the following point clearer in the paper: LAM does *not* introduce additional weights into the model or the learning process. Instead, it simply uses the weights of the classifier as a natural indicator of the importance of each feature dimension to each class. In other words, here the weights are not newly added parameters, but just from the classification head of the original model. So, LAM is not simply built upon a feature matching model. As a matter of fact, the term $w_{uy_k}f_{\phi}^{u}(x_k)$ in the formulation of LAM is exactly the attribution from a feature unit to the logit of certain class in the model. That is why we call it Logit Attribution Matching.

---

> ### Author Response · Authors · 2023-11-16
> **Response to the Reviews (2/2)**
>
> **3. "Before this paper can be accepted, it is necessary that it undergoes revision to ensure a clear and concise explanation of how the theorem serves as a driving force behind the techniques employed."**
>
> Thank you very much for pointing this out. As you kindly suggested, we provide below a clear and concise explanation of why logit attribution matching (LAM) is needed - as a practical implication of the theorem.
>
> In LAM, the labelled pair $(x_k, \tilde{x}_k: y_k)$ is used whereas the unlabelled pair $(x_k, \tilde{x}_k)$ is used in prior CR-based methods. The key here is to realize that the pairing of $x_k$ and $\tilde{x}_k $ in the labelled pair $(x_k, \tilde{x}_k: y_k)$ is *less informative* than the corresponding unlabelled pair $(x_k, \tilde{x}_k)$. In fact, $(x_k, \tilde{x}_k)$ means that $x_k$ and $\tilde{x}_k$ contain the same semantic information about ALL classes, while $(x_k, \tilde{x}_k: y_k)$ means that $x_k$ and $\tilde{x}_k$ contain the same semantic information about ONE PARTICULAR class $y_k$.
>
> This distinction between labelled and unlabelled pairs matters because, in practice, it is much harder to ensure that $x_k$ and $\tilde{x}_k$ hold identical information about all classes of interest than a single class $y_k$, especially when the total number of classes is large. For instance, images labelled with class $y_k$ may include small objects of other classes $y_k'$ in the background. This kind of label noise is pervasive in machine learning datasets [5]. When creating the augmented example $\tilde{x}_k$ from $x_k$, it is much easier to retain information about only $y_k$ since we do not need to care about features of other classes when augmenting the data to randomize non-core features $x^n_k$.
>
> In the aforementioned realistic scenarios, matching the probability distributions over ALL classes is not justified and would lead to sub-optimal solutions (the same can be said about matching the entire feature vectors, $f(x_k)$ and $f(\tilde{x}_k)$). Instead, the logical thing to do is to match only the features that are important for the $y_k$ class. Those considerations lead to LAM. In LAM, regularization is applied only to the labelled pairs $(x_k, \tilde{x}_k: y_k)$, with the assumption that $x_k$ and $\tilde{x}_k$ share the same information of $y_k$. As explained earlier, this assumption is much more practically reasonable since it significantly lowers the requirement on the quality of semantic sharing pairs. The advantage of LAM in this respect is specifically demonstrated through the iWildCam-N dataset with amplified label noise. As shown in Table 1 and discussed in Section 5.4 of the paper, on this dataset, only LAM can improve OOD performance over the ERM+DA baseline (which simply use the augmented examples as additional training data), while all previous CR-based methods that regularize across the entire feature, logit, or probability vector fall short of the baseline.
>
> To summarize, the theorem essentially provides us with a general understanding of optimal DG under ideal conditions where probability matching across all classes works best. However, when practical aspects of the problem are also considered, we must relax the condition. Under the relaxed condition, LAM is the best choice that we currently have in the literature (to the best of our knowledge).
>
> ***
> We plan to appropriately include the points discussed above in the revised version of the paper. Please feel free to let us know if our responses well address your concerns, and feel free to let us know if you have any further questions or concerns. We would be more than pleased to engage in further discussion.
>
> **Reference:**
>
> [1] Zhou, Kaiyang, Ziwei Liu, Yu Qiao, Tao Xiang, and Chen Change Loy. "Domain generalization: A survey." IEEE Transactions on Pattern Analysis and Machine Intelligence (2022).
>
> [2] Wang, Jindong, Cuiling Lan, Chang Liu, Yidong Ouyang, Tao Qin, Wang Lu, Yiqiang Chen, Wenjun Zeng, and Philip Yu. "Generalizing to unseen domains: A survey on domain generalization." IEEE Transactions on Knowledge and Data Engineering (2022).
>
> [3] Irena Gao, Shiori Sagawa, Pang Wei Koh, Tatsunori Hashimoto, Percy Liang; Proceedings of the 40th International Conference on Machine Learning, PMLR 202:10800-10834.
>
> [4] Peters, Jonas, Peter Bühlmann, and Nicolai Meinshausen. "Causal inference by using invariant prediction: identification and confidence intervals." Journal of the Royal Statistical Society Series B: Statistical Methodology 78.5 (2016): 947-1012.
>
> [5] Northcutt, Curtis G., Anish Athalye, and Jonas Mueller. "Pervasive Label Errors in Test Sets Destabilize Machine Learning Benchmarks." In Thirty-fifth Conference on Neural Information Processing Systems Datasets and Benchmarks Track. 2021.

---

### Author Response · Authors · 2023-11-23
**Revisions and Updates to the Manuscript**

We deeply appreciate all reviewers for your meticulous reviews with our work. Based on your feedback and subsequent discussions, we have revised the manuscript accordingly. All updates are highlighted in blue. Below, we outline the key changes:

1.	As Reviewer 4UiZ and 9AoB observed, the link between the theorem and our proposed LAM wasn't sufficiently discussed. To address this, we've expanded the discussion in Section 4.2.
2.	Reviewer H9LW and 4UiZ expressed confusion about the weights in our proposed LAM regularization term. We apologize for the lack of clarity. To rectify this, we've added a sentence right after the introduction of the LAM regularization term in Section 4.2, clarifying that $w_{uy_k}$ represents the weight in the classification layer between feature unit $u$ and the logit of class $y_k$.
3.	Reviewer THPX, cViE, and 9AoB inquired about how LAM would interact with more generic data augmentation methods. Initially, we did include the results of LAM and other CR-based DG methods on iWildCam and iWildCam-N with RandAugment used in Table 8 (Appendix E). Now, we've also incorporated the results for Camelyon in Table 8. As suggested by Reviewer THPX, we've added a sentence in Section 5.4 highlighting the superior performance of LAM when RandAugment, a more generic data augmentation method, is used. This demonstrates the wide applicability of LAM with various DA methods.
4.	Reviewer 9AoB expressed confusion about the contribution of CR on top of DA. We apologize for the misunderstanding. To clarify, we've revised the introduction section emphasizing that our work doesn't claim to be the first to propose CR methods for DG. Rather, we are highlighting that our work is among the first to provide an extensive analysis and comparison of existing CR methods for DG under the same evaluation protocol and dataset.


Due to time constraints, we were unable to address every point raised by the reviewers in this revision, and chose to focus on the major issues identified by multiple reviewers. However, we commit to thoroughly considering and integrating all feedback in the forthcoming version of this paper, regardless of whether it is for ICLR 2024 or other conferences.

We are grateful for your insightful comments, which have significantly contributed to the improvement of our manuscript. We look forward to further feedback and discussions.

---

### Public Comment · ~Weiyan_Xie1 · 2024-07-10
**Updated Version of Our Paper Accepted at UAI 2024**

An updated version of our paper has been accepted at UAI 2024. You can access the revised manuscript [here](https://openreview.net/forum?id=WNy1ooHYHx&referrer=%5Bthe%20profile%20of%20Nevin%20L.%20Zhang%5D(%2Fprofile%3Fid%3D~Nevin_L._Zhang1).

Best,

Authors

---

### Meta-Review · Area_Chair_JRdr · 2023-12-04

**Metareview:**

The paper provides a consistency regularization technique for domain generalization. The writing was praised by the reviewers, who found the paper accessible. Additionally, the casual connection was interesting. The paper also provides a theoretical statement, showing that consistency regularization is optimal for domain generalization, but the specific connection to this method is unclear. Overall, the empirical improvement is marginal, and the method alone is not sufficiently novel. I think that a clearer connection between the theory and the method would help strengthen the paper for resubmission. If a strong connection is not there, I would recommend the authors strengthen and pursue one of the two directions (theoretical understanding of consistency regularization or methodological improvements).

**Justification For Why Not Higher Score:**

The theoretical result is disconnected from the method. Neither of the two is sufficient for acceptance on its own at this stage.

**Justification For Why Not Lower Score:**

N/A

---

### Decision · Program_Chairs · 2024-01-16

Reject